# Identification of host–pathogen-disease relationships using a scalable multiplex serology platform in UK Biobank

Alexander J. Mentzer [1,2,31✉], Nicole Brenner [3,31], Naomi Allen [2,4,5], Thomas J. Littlejohns [2,5], Amanda Y. Chong [1], Adrian Cortes[2], Rachael Almond[4], Michael Hill[5,6], Simon Sheard [4], Gil McVean [2], UKB Infection Advisory Board*, Rory Collins[4,5], Adrian V. S. Hill[1,7] & Tim Waterboer [3✉]

Certain infectious agents are recognised causes of cancer and other chronic diseases. To understand the pathological mechanisms underlying such relationships, here we design a Multiplex Serology platform to measure quantitative antibody responses against 45 antigens from 20 infectious agents including human herpes, hepatitis, polyoma, papilloma, and ret-roviruses, as well as *Chlamydia trachomatis*, *Helicobacter pylori* and *Toxoplasma gondii*, then assayed a random subset of 9695 UK Biobank participants. We find seroprevalence esti-mates consistent with those expected from prior literature and confirm multiple associations of antibody responses with sociodemographic characteristics (e.g., lifetime sexual partners with *C. trachomatis*), *HLA* genetic variants (rs6927022 with Epstein-Barr virus (EBV) EBNA1 antibodies) and disease outcomes (human papillomavirus-16 seropositivity with cervical intraepithelial neoplasia, and EBV responses with multiple sclerosis). Our accessible dataset is one of the largest incorporating diverse infectious agents in a prospective UK cohort offering opportunities to improve our understanding of host-pathogen-disease relationships with significant clinical and public health implications.

[1] The Wellcome Centre for Human Genetics, University of Oxford, Oxford, UK. [2] Big Data Institute, Li Ka Shing Centre for Health Information and Discovery, University of Oxford, Oxford, UK. [3] Division of Infections and Cancer Epidemiology, German Cancer Research Center (DKFZ), Heidelberg, Germany. [4] UK Biobank, Stockport, UK. [5] Nuffield Department of Population Health, University of Oxford, Oxford, UK. [6] MRC-Population Health Research Unit, University of Oxford, Oxford, UK. [7] The Jenner Institute, University of Oxford, Oxford, UK. [31]These authors contributed equally: Alexander J Mentzer, Nicole Brenner. *A list of authors and their affiliations appears at the end of the paper. ✉email: alexander.mentzer@ndm.ox.ac.uk; T.Waterboer@dkfz-heidelberg.de

Infectious microbes are well-established causal agents in the development of some non-communicable diseases (NCDs) and are strongly suspected to contribute to the development of many other diseases afflicting humans. For example, the International Agency for Research on Cancer (IARC) have listed ten microbes as being well-established (Group 1) carcinogenic agents. These include *Helicobacter pylori* (Hp), hepatitis B (HBV) and hepatitis C (HCV) viruses, multiple types of human papillomavirus (HPV), Epstein-Barr virus (EBV) and Kaposi's sarcoma-associated herpesvirus (KSHV; human herpesvirus 8), as well as parasitic agents endemic in the tropics all associated with particular subtypes of cancer[1]. In many of these examples the molecular mechanisms underlying cancer development is well understood, such as with HPV-16 causing cervical cancer[2] and EBV with Burkitt's lymphoma[3]. For most other IARC-reported associations the fine detail of mechanisms remain obscure. Furthermore, even in circumstances where molecular mechanisms are better understood for specific infection-cancer outcomes, such processes cannot be directly extrapolated to other forms of cancer caused by the same agent (e.g., cervical and oropharyngeal cancer for HPV-16, or nasopharyngeal and gastric cancer for EBV). There is also the potential that these infectious agents may contribute to other rarer cancer types and that the detection of such association signals may be more difficult to identify in standard epidemiological study formats. Another challenge of understanding any infection-cancer association, is the observation that exposure to such infectious agents do not confer cancer on every exposed individual suggesting the presence of significant risk modifying factors that may be numerous and varied across individuals.

Moreover, the existing IARC list is not exhaustive and cancer is not the only long-term sequelae of infectious agents. A number of microbes not formally listed by IARC have been putatively associated with the development of diseases including cancer, atherosclerosis or inflammatory conditions such as multiple sclerosis. Cytomegalovirus (CMV), for example is another human herpesvirus that is acquired later in adolescent or adult life and has been linked with the development of atherosclerosis and subsequent cardiovascular events such as myocardial infarction[4]. Furthermore, there is extensive literature linking EBV with multiple sclerosis[5–7]. However, much of the evidence supporting these associations has been derived from cross-sectional or case-control analyses that are unable to determine temporality, or from small nested case-control studies that have generally been small or have focused on a small numbers of infectious agents, leading to inconsistent findings[8,9]. Definitive evidence to support or refute these postulated associations are most likely to come from prospective cohort studies. In such studies it is possible to measure potential exposures (be they infectious, lifestyle or inherent to the individual such as genetic) in a sample of the population at baseline recruitment and then follow up individuals until they develop disease to thus define exposure and disease temporality.

UK Biobank (UKB) was designed as a large prospective cohort study that has collected a substantial amount of genetic, lifestyle, and biomarker data alongside a wide range of health outcomes with linkage still ongoing for over 500,000 adults in the United Kingdom[10]. The depth and breadth of data available in UKB alongside repeated biological sampling in random subsets every few years offer unique opportunities to explore the interrelationships between risk factors and disease outcomes in very large numbers of individuals. The availability of serological data characterising the exposure history for multiple infectious agents for a large number of adults in middle to older age would allow the retesting of putative and identification of novel associations between infectious agents and multiple disease outcomes whilst accounting for potential confounders. Serial follow up measures even in small numbers of individuals may facilitate an improved understanding of changing exposure risks over time in the cohort. Moreover, since many infectious agents such as EBV, CMV and human immunodeficiency virus (HIV-1) are considered to have modulatory effects on the immune system[11,12], it is equally important to consider their exposure history when testing other non-infectious exposure and disease risk associations. Understanding the role of infectious agents in the development of NCDs could have major implications for guiding public health decisions such as targeted vaccination strategies for primary prevention[13].

Measuring exposure to infectious agents is most frequently undertaken by measuring antibodies that, for many agents, are stable makers of exposure that may change in magnitude based on high or low viral loads, or following repeated exposure or reactivation such as with *Hp* or varicella-zoster virus (VZV) respectively[14]. Antibody assays are simple to perform but in their traditional solid phase format, it rapidly becomes impractical to perform assays for measuring antibodies against multiple pathogen-specific antigens in large studies. Recent technological advances have facilitated the development of high throughput systems capable of measuring antibodies against multiple antigens simultaneously. Flow-based methods incorporating fluorescently labelled synthetic beads with conjugated antigens offer a particularly attractive means to achieve scalability with minimal impact on assay performance. Furthermore, such methods permit the testing for antibodies against multiple antigens and pathogens simultaneously which may then enable a deeper understanding of how specific antigen exposure may differentially associate with disease susceptibility. In particular, the suspension array technology commonly referred to as "Multiplex Serology"[15] developed at the German Cancer Research Center has been successfully utilised in multiple large seroepidemiological investigations[16,17]. The application of such technology in UKB offers the opportunity to test for the risk of prior exposure to infectious agents (through cross-sectional antibody measure), and subsequent incidence of cases of a disease of interest, adjusting for potential confounders that should provide a more reliable estimation of risk and causality.

Here, we describe the results from the first phase of a project to measure antibody responses against 45 antigens from 20 infectious agents using an extended and fully validated Multiplex Serology panel that we intend to apply to all 500,000 UKB participants. Using a randomly selected subset of 9695 individuals, we demonstrate the high performance characteristics of the platform specifically adapted for high throughput and demonstrate the compatibility with prior single agent studies through confirming expected seroprevalence estimates and reproducing previously reported cross-sectional epidemiological and genetic associations with infectious agent exposure that may have utility in a range of future explorations. We highlight the significant benefit of these Multiplex Serology data when used alongside existing rich UKB data and their potential utility for understanding NCD development, with particular relevance to two exemplar diseases: cervical cancer and multiple sclerosis (MS).

## Results

**UK biobank sample overview**. The baseline demographic and behavioural characteristics of the 9695 UKB individuals selected at random and included in this Multiplex Serology study were comparable with the remaining 493,852 individuals with serum available for similar future analyses (Table 1). Consistent with previous reports from UKB[18], this sample consists of a greater proportion of females (55.9%) versus males, individuals of self-reported White ethnicity (94.3%) and those of a more affluent

**Table 1 Baseline characteristics of the 9,695 randomly selected individuals with Multiplex Serology data compared with the remainder of the UKB cohort with equivalent samples remaining for future analyses.**

| Characteristic (range where applicable) | Multiplex serology measures, Numbers (%) N = 9,695 | Remaining UKB cohort, Numbers (%)* N = 493,852 |
|---|---|---|
| **Age at recruitment** | | |
| 40–49 | 2543 (26.2) | 129,899 (26.3) |
| 50–59 | 3390 (35.0) | 174,181 (35.3) |
| 60–69 | 3762 (38.8) | 189,765 (38.4) |
| **Sex** | | |
| Male | 4271 (44.1) | 225,064 (45.6) |
| Female | 5424 (55.9) | 268,788 (54.4) |
| **Ethnic group** | | |
| White | 9140 (94.3) | 464,606 (94.1) |
| Asian | 236 (2.4) | 11238 (2.3) |
| Black | 141 (1.5) | 7877 (1.6) |
| Other | 134 (1.4) | 7390 (1.5) |
| Not reported/ missing | 44 (0.4) | 2741 (0.5) |
| **Townsend deprivation index†** | | |
| Less than −2 (more affluent) | 5106 (52.7) | 259,869 (51.7) |
| −2 to 2 (average) | 3046 (31.4) | 159,450 (31.7) |
| Greater than 2 (more deprived) | 1533 (15.8) | 82343 (16.4) |
| Not reported/ missing | 10 (0.1) | 881 (0.2) |
| **Smoking status** | | |
| Never | 5377 (55.5) | 268,842 (54.5) |
| Previous | 3308 (34.1) | 170,052 (34.4) |
| Current | 956 (9.8) | 52048 (10.5) |
| Not reported/ missing | 54 (0.6) | 2910 (0.6) |
| **Alcohol drinker status** | | |
| Never | 425 (4.4) | 21976 (4.4) |
| Previous | 338 (3.5) | 17819 (3.6) |
| Current | 8913 (91.9) | 452,424 (91.7) |
| Not reported/ missing | 19 (0.2) | 1633 (0.3) |
| **Number of sexual partners** | | |
| 0 | 89 (0.9) | 4148 (0.8) |
| 1 | 2338 (24.1) | 114,636 (23.2) |
| 2–4 | 2534 (26.1) | 128,428 (26.0) |
| 5–10 | 2058 (21.2) | 103,251 (20.9) |
| > 10 | 1007 (10.4) | 51751 (10.5) |
| Not reported/ missing | 1669 (17.3) | 91638 (18.6) |
| **Ever had same sex intercourse** | | |
| Yes | 295 (3.0) | 15819 (3.1) |
| No | 8442 (87,1) | 437,256 (87.0) |
| Not reported/ missing | 958 (9.9) | 49468 (9.9) |

*UKB participants with >100 μL serum.
†Townsend deprivation index presented here in categories of integers for representation purposes but modelled in quintiles.

or perceived risk of NCD development (described further in the Supplementary Methods). Following a consensus selection (Supplementary Table 1), Multiplex Serology was used to measure serum antibody levels against 20 infectious agents (Supplementary Table 2) including human herpesviruses 1-8; HIV-1; human T-lymphotropic virus-1 (HTLV-1); HBV and HCV; HPV-16 and -18; the JC (JCV), BK (BKV) and Merkel cell (MCV) polyomaviruses; *Hp*; *Chlamydia trachomatis* (*Ct*); and *Toxoplasma gondii* (*Tg*). The Multiplex Serology methodology was developed 15 years ago for HPV and has since been extended to numerous other infectious agents and has been applied to multiple seroepidemiological studies[15,19]. We applied a systematic process of validation for tailored panel and application of Multiplex Serology to the UKB sample set (outlined in Supplementary Fig. 1) with details described in the Methods, Supplementary Methods and Supplementary Fig. 2. The major difference of the assay process used in this study compared to previously published Multiplex Serology studies is the use of magnetic instead of non-magnetic beads to allow for automation of previously manually handled steps and therefore enhancing scalability. To validate the performance of the assay using magnetic compared to non-magnetic beads, we assessed agreement in assay metrics between the different bead sets and observed a high level of agreement for all infectious agents (median intra-class correlation coefficient (ICC) 0.94, Supplementary Table 4) except for *Tg* which was an outlier with an ICC estimate of 0.48. This was likely due to lower specificity but higher sensitivity on non-magnetic versus magnetic beads (Supplementary Fig. 3).

Following completion of Multiplex Serology validation, the UKB samples were tested on six assay days spaced over two calendar weeks with more detail on the specific steps provided in the Supplementary Methods. The possibility of differences by batch were tested for by measuring the correlation between MFI values for all antigens using 'bridging panel samples' (standard samples run in duplicate across the runs). An example of the levels of correlation observed between days 1 and 3 are shown in Supplementary Fig. 4 where correlation was found to be high (median $r^2$ = 0.95, rho = 0.96). Estimates for all comparisons for all antigens across all 6 testing days are provided in Supplementary Table 4 and all estimates are in keeping with minimal batch variation across the testing days. The tested UKB serum samples also included interspersed blind-spiked duplicates for 107 (1.1%) individuals to further quantify within- and between-batch variation which was found to be low (median coefficient of variation (CV) among all samples 17%, and 3.5% among seropositives only; see Supplementary Table 5 and Supplementary Methods for the rationale of testing only seropositives). Furthermore, inter-plate variance was calculated between plates as determined within a day, or between days, using three control samples tested on every plate. The calculated CV for all control samples and antigens were again found to be low (median across all antigens 16% within days and 19% between days; Supplementary Fig. 5). Consistency of serostatus between baseline and repeat assessment samples were also tested for 277 individuals to estimate seroconversion and seroreversion rates (Supplementary Methods). Seroconversion rates within the 3-5 year interval between baseline and repeat assessments ranged from 0–10.5% but seroreversion rates of 0–9.4% were also observed (Supplementary Table 6).

background (52.7 with a Townsend deprivation index (TDI) less than -2). There was a high proportion of individuals who consumed alcohol (91.9%) with only 9.8% of individuals reporting current smoking.

**Multiplex serology applied to the UKB sample**. A Working Group was convened to make recommendations on the most relevant infectious agents to screen for in the UKB with relevance to public health in the UK, particularly considering the confirmed

**Associations between demographic factors and infectious agent seropositivity**. Using defined algorithms to estimate the seroprevalence of specific infectious agents using results from, in some cases, multiple antigens per agent (as described in detail in the Supplementary Methods and for HHV-6 and HHV-7 in

**Table 2 Unadjusted seroprevalence estimates and 95% confidence intervals for infectious agents tested using the Multiplex Serology platform in all 9695 individuals, overall and stratified by sex.**

| Infectious Agent | Overall prevalence (%, 95% CI) | Seroprevalence males (%, 95% CI) | Seroprevalence females (%, 95% CI) | Expected prevalence (range in %) |
|---|---|---|---|---|
| HSV-1 | 69.8 (68.9–70.7) | 68.7 (67.3–70.1) | 70.7 (69.5–72.0) | 60-80[29,48] |
| HSV-2 | 16.2 (15.5–16.9) | 15.3 (14.2–16.4) | 16.9 (15.9–17.9) | 2-25[29,49] |
| VZV | 92.5 (92.0–93.0) | **94 (93.2–94.7)** | **91.4 (90.6–92.1)** | 87-97[50–52] |
| EBV | 94.7 (94.3–95.2) | **93.4 (92.7–94.2)** | **95.7 (95.2–96.2)** | 80-98[52–54] |
| CMV | 58.2 (57.2–59.2) | 56.8 (55.3–58.3) | 59.3 (58.0–60.7) | 40-70[28,52,55,56] |
| HHV-6A or 6B | 90.8 (90.2–91.4) | 91 (90.1–91.8) | 90.7 (89.9–91.5) | >90*[57,58] |
| HHV-7 | 94.7 (94.3–95.2) | **92.9 (92.1–93.7)** | **96.2 (95.7–96.7)** | >85*[59,60] |
| KSHV | 8.1 (7.5–8.6) | 8.7 (7.9–9.6) | 7.6 (6.9–8.3) | 1.5-12[61–63] |
| HBV | 2.5 (2.2–2.8) | **3.3 (2.9–3.9)** | **1.8 (1.4–2.1)** | 0.1-3[64–66] |
| HCV | 0.3 (0.2–0.4) | 0.4 (0.2–0.6) | 0.2 (0.1–0.3) | 0.01-1[64,65,67,68] |
| HIV-1 | 0.2 (0.1–0.3) | 0.3 (0.1–0.5) | 0.1 (0.1–0.2) | 0.05-0.2[64,69,70] |
| HTLV-1 | 1.6 (1.3–1.8) | **2.0 (1.6–2.4)** | **1.0 (0.9–1.5)** | 0.003-1[71,72] |
| HPV-16† | 4.4 (4.2–4.8) | **2.7 (2.2–3.2)** | **5.7 (5.1–6.4)** | 3-12[73–75] |
| HPV-18† | 2.8 (2.4–3.1) | **1.9 (1.5–2.4)** | **3.4 (2.9–3.8)** | 1-8[73,74] |
| JCV | 57.5 (56.5–58.5) | **60.9 (59.4–62.4)** | **54.8 (53.5–56.2)** | 45-65[76–78] |
| BKV | 95.4 (95.0–95.8) | **96.1 (95.5–96.7)** | **94.8 (94.2–95.4)** | 90-99[76,78] |
| MCV | 66.7 (65.8–67.7) | 67.7 (66.3–69.1) | 66 (64.7–67.3) | 60-80[78–80] |
| *T. gondii* | 28 (27.1–28.9) | 29.0 (27.6–30.4) | 27.2 (26.0–28.4) | 7-40[30,81] |
| *C. trachomatis* | 21.4 (20.6–22.2) | **15.9 (14.8–17.0)** | **25.7 (24.5–26.9)** | 13-25[26] |
| *H. pylori*‡ | 35.3 (34.0–36.6) | **38.2 (36.1–40.3)** | **33.1 (31.4–34.9)** | 13-50[82] |

*Serological detection methods for HHV-6 family members are prone to cross-reactivity with HHV-7 and other viruses[58] and therefore our estimates should be considered provisional although we observed very little correlation between HHV-6 and HHV-7 antigen responses ($r < 0.2$, Supplementary Methods and Supplementary Fig. 4).
†Prior published estimates for human papillomavirus are based on a combination of cervical DNA detection which will be dependent on either acute infection or reactivation; or some limited seroprevalence studies in population controls (as opposed to cancer case cohorts).
‡Estimates calculated from 50% of samples (see Supplementary Methods for details).
Estimates in men and women highlighted in bold are significantly different from each other (i.e. they do not have overlapping confidence intervals). Ranges of previously published seroprevalence estimates representing expectations are provided as a reference.

Supplementary Fig. 6), we found our estimates to be within the estimate ranges reported in the literature for equivalent British or, if not available, other European or worldwide populations (Table 2. For example, we confirmed near-ubiquitous exposure to VZV (92.5%, 95% C.I. 92.0–93.0%), EBV (94.7%, 94.3–95.2%) and BKV (95.4%, 95.0–95.8%), and rare exposure against the blood-borne viruses HIV-1 (0.2%, 0.1–0.3%), HBV (2.5%, 2.2–2.8%) and HCV (0.3%, 0.2–0.4%).

We observed significant differences in seroprevalence estimates between males and females for 11 infectious agents (Table 2), with five found to be higher in women (EBV, HHV-7, HPV-16, HPV-18 and *Ct*) and six higher in men (VZV, HBV, HTLV-1, JCV and BKV and *Hp*). For all of the agents where data from multiple antigens were used to calculate total infectious agent seroprevalences (EBV, HBV, HTLV-1, *Hp*), equivalent differences were observed in the same directions for the individual pathogen-specific antigens (Supplementary Table 7). The differences observed in crude seroprevalences by sex for these 11 infectious agents persisted following adjustment for age, TDI, self-reported ethnicity, LSP and a report of ever having had sexual intercourse with a member of the same sex (sameSI; Supplementary Table 8). We also observed significant differences in seroprevalence by age (Fig. 1A and B and Supplementary Table 9 presenting all infectious agents with significant demographic associations after adjustment), being most significant for HSV-1, *Tg* and notably CMV where 49.2% (95% C.I. 46.6–51.8%) of women between the ages of 40 and 50 were seropositive compared to 66.8% (64.8–68.9%) of women between the ages of 60 and 70. Age was inversely associated with HPV-16, JCV and BKV. For example, BKV seroprevalence was 97.4% (96.5–98.4%) in men aged 40–49 years, and 95.1% (94.1–96.1%) in 60–69 year olds.

We used the limited number of serial samples available (277 individuals, median time between sampling 4.6 years, range 2.4–6.0) to test whether the observed increased seroprevalence

estimates for particular infectious agents found associated with age indicated continued exposure to the infectious agent over the course of UKB follow-up. Using a linear mixed model adjusting for TDI, LSP and ethnicity as fixed effect covariates and individual as a random effect covariate we observed a significant increase in $\log_{10}$ transformed antibody levels over time for all three CMV antigens (beta 0.05, SE 0.02, $P$ 0.03 with pp28; beta 0.08, SE 0.03, $P$ 0.004 with pp52; and beta 0.11, SE 0.03, $P$ 0.0004 with pp150N) and one *Tg* antigen (beta 0.06, SE 0.03, $P$ 0.03 for p22) but not for the HSV-1 gG antigen. We also observed an increase in likelihood of seroprevalence over time for CMV (beta 0.04, SE 0.02, $P$ 0.02) and *Tg* (beta 0.08, SE 0.03, $P$ 0.01) but again not for HSV-1. Thus these data support ongoing infectious exposure over the ages of UKB participation for those sampled repeatedly for *Tg* and CMV, but not for HSV-1. Conversely, for VZV, where we expected to see changes more consistent with reactivation of virus with increasing age of individuals during the ongoing participation of individuals in UKB, we observed a significant increase in VZV antibody levels over time (beta 0.05, SE 0.02, $P = 0.03$), but not with overall seropositivity in individuals.

We also observed significant differences in seroprevalences of multiple infectious agents stratified by self-reported ethnicity (Fig. 1C and D and Supplementary Table 10). For example, there was a higher CMV seroprevalence in Asian women (90.4%; 84.9–95.8%) and men (93.4%, 89.1–97.1%) compared to White women (57.8%; 56.5–59.2%) and men (54.8%, 53.2–56.3%), and a higher HBV seroprevalence in Black women (14.5%; 6.9–22.0%) and men (36.2%, 23.8–48.6%) compared to White women (1.2%; 0.9–1.5%) and men (2.2%, 1.8–2.7%). We also found a higher seroprevalence of several infectious agents known to be transmitted through sexual or close physical contact with increasing LSP (HSV-1, HSV-2, EBV, HPV-16 and HPV-18 and *Ct*; Fig. 1E and F and Supplementary Table 11). We did not observe significant

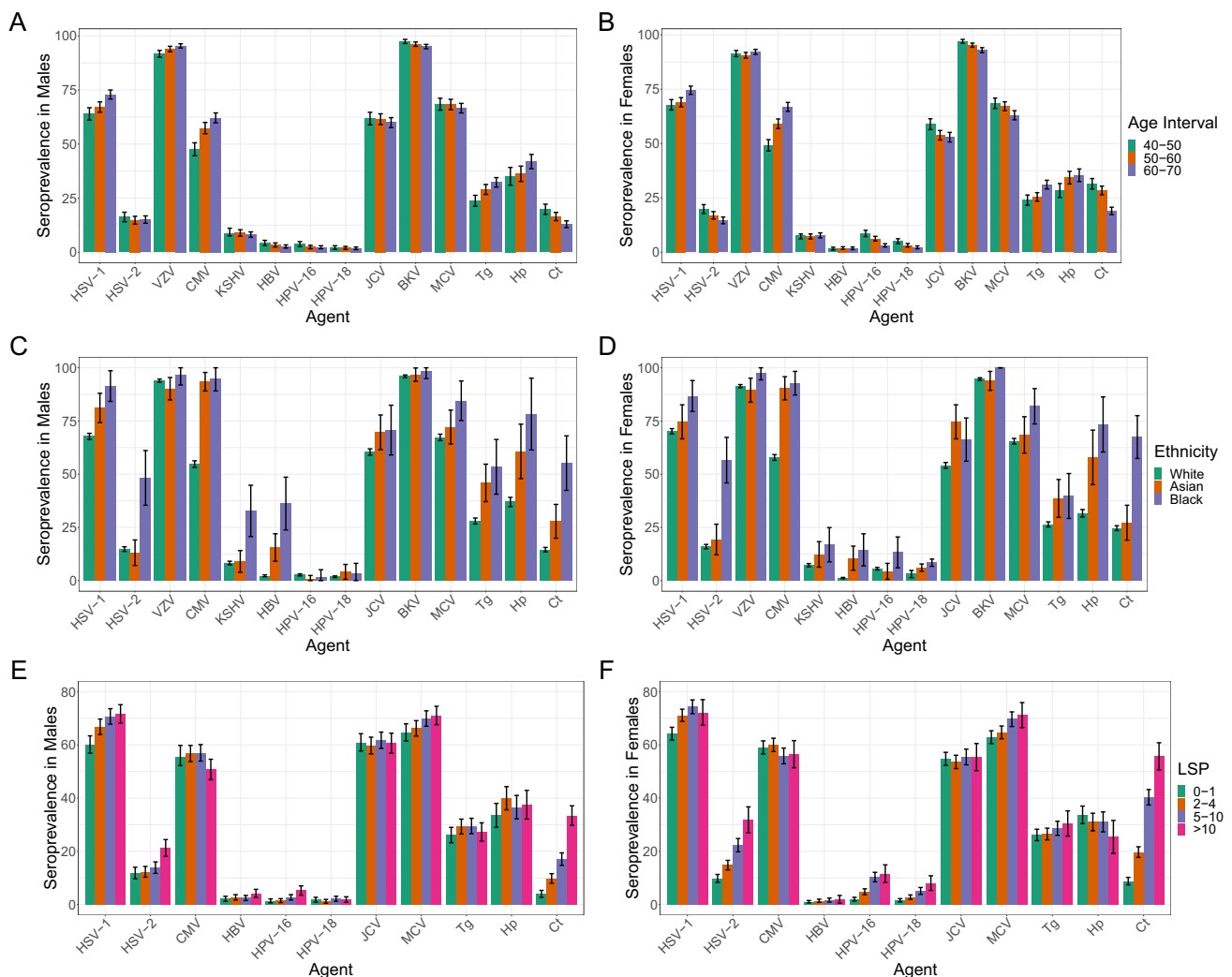

**Fig. 1 Crude seroprevalence estimates and 95% confidence intervals for multiple infectious agents in 9695 biologically independent UK Biobank participants stratified by sex and age (A for males and B for females), self-reported ethnicity (C for males and D for females) and LSP (E for males and F for females).** Precise numbers of individuals in each category are provided in Supplementary Tables 9–11. The infectious agents represented are those with significant ($P < 0.01$) evidence of differences based on crude estimates when testing for association with either males or females alone, or combined with each of age, self-reported ethnicity and LSP where sufficient numbers of individuals were available in each group for models to converge. Evidence of interaction with sex was only observed for VZV ($P = 0.01$) and HPV-16 ($P = 0.04$) with age, and HSV-2 ($P = 7.5 \times 10^{-5}$) and HPV-18 ($P = 0.01$) with LSP. All $P$-values reported were from the likelihood ratio test with no adjustment for multiple testing.

**Table 3 Crude seroprevalence and adjusted odds ratios for HBV, HIV-1 and *Ct* in men reporting same sex intercourse in males.**

| Infectious agent | Ever had same sex intercourse | Numbers | Seroprevalence (95%CI) | Adjusted OR (95%CI) | *P*-value |
|---|---|---|---|---|---|
| **HBV** | No | 3702 | 2.8 (2.2–3.3) | 1.0 | |
| | Yes | 176 | 13.6 (8.6–18.7) | 3.8 (1.9–7.7) | $1.4 \times 10^{-3}$ |
| **HIV-1** | No | 3702 | 0.1 (0.1–0.3) | 1.0 | |
| | Yes | 176 | 4.0 (1.1–6.9) | 10.8 (1.9–59.1) | $6.2 \times 10^{-3}$ |
| ***Ct*** | No | 3702 | 15.1 (13.9–16.2) | 1.0 | |
| | Yes | 176 | 33.5 (26.6–40.5) | 1.73 (1.15–2.59) | $8.7 \times 10^{-3}$ |

Odds ratios adjusted for age interval, ethnicity, TDI and number of reported lifetime sexual partners with *P*-values reported using logistic regression.

associations between LSP and rare sexually transmitted infections such as HIV-1, HBV, HCV or HTLV-1. However, we did observe statistically significant associations in men reporting sameSI for two of these infections (HBV, HIV-1) as well as *Ct* (Table 3). We observed an association between higher socio-economic deprivation and many infectious agents including HSV-1, HSV-2, EBV, CMV, KSHV, HBV, *Tg*, *Hp* and *Ct* (Supplementary Table 12). This

was particularly striking for *Hp* in women where 25.7% (22.2–29.2%) of women in the least deprived group had evidence of exposure compared to 42.8% (38.6–47.0%) in the most deprived group ($P_{trend} = 4.3 \times 10^{-11}$). Similarly, 19.7% (17.4–22.1%) of women in the l least deprived group had evidence of exposure to *Ct* compared to 34.8% (31.9–37.7%) of women in the most deprived ($P_{trend} = 2.5 \times 10^{-16}$). The same trends were also

**Table 4 Replication of genetic variants previously associated with exposure to specific infectious agent antigens.**

| Infectious agent antigen | Reported associated variant | Trait description* | Previously reported OR (95% CI)† | Previously reported P-value | UKB beta | UKB OR‡ | UKB P-value□ |
|---|---|---|---|---|---|---|---|
| JCV VP1 | rs9269910 | Seropositivity | 1.74 (1.58–1.90) | $8.9 \times 10^{-12}$ | 0.12 | 1.65 | $6.9 \times 10^{-22}$ |
| MCV VP1 | rs9269268 | Seropositivity | 1.53 (1.40–1.66) | $2.67 \times 10^{-10}$ | 0.10 | 2.82 | $7.4 \times 10^{-32}$ |
| EBV EBNA1 | rs6927022 | Quantitative | 1.17 (1.14–1.21) | $7.35 \times 10^{-26}$ | 0.29 | 3.31 | $9.5 \times 10^{-91}$ |

*Seropositivity refers to a binary comparison between seropositive and seronegative individuals. Quantitative refers to a linear test of association using log(10) transformed MFI values.
†Odds ratios for quantitative responses were calculated by transforming reported beta estimates through the natural logarithm.
‡Odds Ratios (ORs) only available from beta and standard error statistics determined using linear mixed model genetic association.
□ P-values calculated using linear regression from beta coefficients—further details on association testing provided in the Supplementary Methods.

observed for males. However, with the exception of *Hp*, none of the associations remained significant after adjusting for age, ethnic group and LSP.

**Genetic associations with infectious agent antibody responses.** Multiple genetic associations have been discovered to affect magnitude of response or likelihood of seropositivity against several infectious antigens. Of the individuals included in the UKB sample, 9611 also had genetic data available for analyses following quality control. Using this genetic data applied to all individuals and explicitly accounting for population structure and relatedness using linear mixed models we replicated three previously reported associations between human genetic variants and antibody responses against infectious agents (Supplementary Table 13). All three associations were variants within the class II region of the MHC locus of the human genome. Two variants (rs9269910 and rs9269268) were found to be associated with seropositivity for JCV ($P = 6.9 \times 10^{-22}$) and MCV ($P = 1.4 \times 10^{-38}$) respectively, and another variant (rs6927022) was associated with antibody levels against EBV EBNA1 antigen ($P = 9.5 \times 10^{-91}$; Table 4). We found the genetic association signals for JCV and MCV to be more significant when described as continuous traits ($P = 5.0 \times 10^{-63}$ and $1.1 \times 10^{-38}$ respectively). In order to assess the influence of human genetics on responses against the capsid of EBV we tested for associations with the viral capsid antigen (VCAp18) and discovered a statistically significant signal of association between MHC variants (with the most significant association with the rs7197 variant; beta 0.04; 95% CI 0.03–0.05; $P = 1.7 \times 10^{-22}$, Supplementary Fig. 7). All these reported associations remain highly significant even after accounting for multiple testing using Bonferroni correction and a stringent GWAS P-value threshold (nominally $P = 5 \times 10^{-9}$ adjusted to $1 \times 10^{-9}$ for five individual GWAS). We observed that these SNP associations were substantially more significant than imputed HLA allele associations with the same continuous traits (Supplementary Table 14).

**Associations between infectious agent exposure and health outcomes.** In order to demonstrate the utility of the UKB serology dataset for testing associations between infectious agent exposure and chronic disease we tested for cross-sectional association between two sets of infections and diseases. We replicated the well-established association between HPV-16 L1 seropositivity (considered a marker for cumulative exposure) and the risk of cervical intraepithelial neoplasia (CIN; based on 131 cases; OR = 2.65; 95% CI 1.51–4.66, $P = 0.001$) and cervical cancer (18 cases, OR = 4.37; 95% CI 1.26–15.15; $P = 0.04$). The association with CIN remained significant after adjustment for age, ethnicity, LSP and TDI (OR = 2.02; 95% CI 1.06–3.05; $P = 0.03$; Fig. 2). Similarly, we found that all self-reported cases of MS disease in the UKB subset ($n = 34$) were seropositive for EBV infection as determined using either our validated algorithm incorporating all four EBV antigens, or using VCAp18 seropositivity alone (Fig. 3A) which remained significant after adjustment for age, sex, ethnic group and TDI (OR = 5.3; 1.55–18.39; $P = 7.8 \times 10^{-3}$). A similar, non-significant pattern of

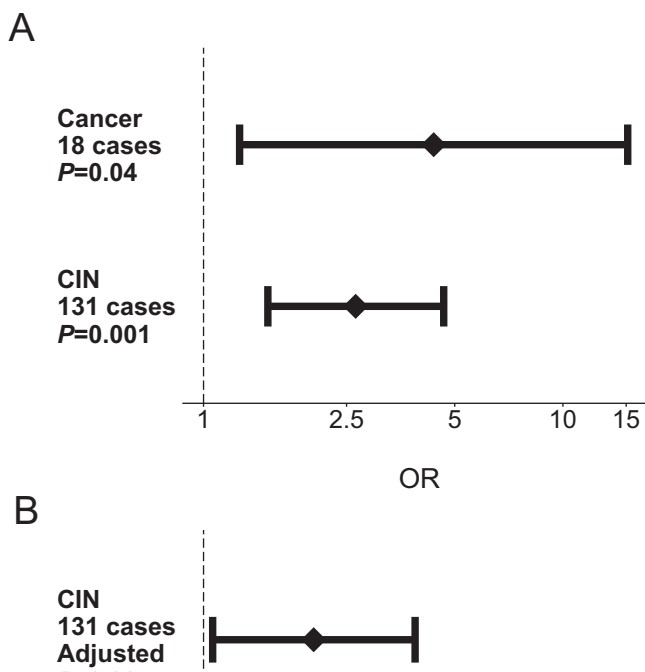

**Fig. 2 Association between HPV-16 L1 antigen seropositivity and risk of cervical cancer and cervical intraepithelial neoplasia (CIN) in 9695 UK Biobank participants. A** The diamonds represent unadjusted odds ratios of cervical cancer and cervical intraepithelial neoplasia by HPV-16 L1 seropositivity, shown with 95% confidence interval error bars (P-values calculated with two-sided Fisher's exact test). **B** Odds ratio and 95% confidence interval of CIN risk after adjustment for age, sex, ethnicity, LSP and TDI (P-value calculated using multivariable regression analysis).

association was observed for VCAp18 antibody levels (Fig. 3B). We also observed an inverse association with CMV seroprevalence and risk of MS that persisted after adjustment for the same covariates (OR = 0.39; 0.18–0.81; $P = 0.01$).

**Genetic correlation analysis.** Since we had limited power within the UKB to demonstrate a robust association between EBV exposure and risk of MS we used the genetic data available in UKB to test for correlation in genetic architecture between the two traits. Using imputed HLA allele information in European ancestry individuals within UKB, we found that those alleles associated with MS disease risk (as reported in the largest HLA GWAS of MS to date[20]) had similar statistics of association

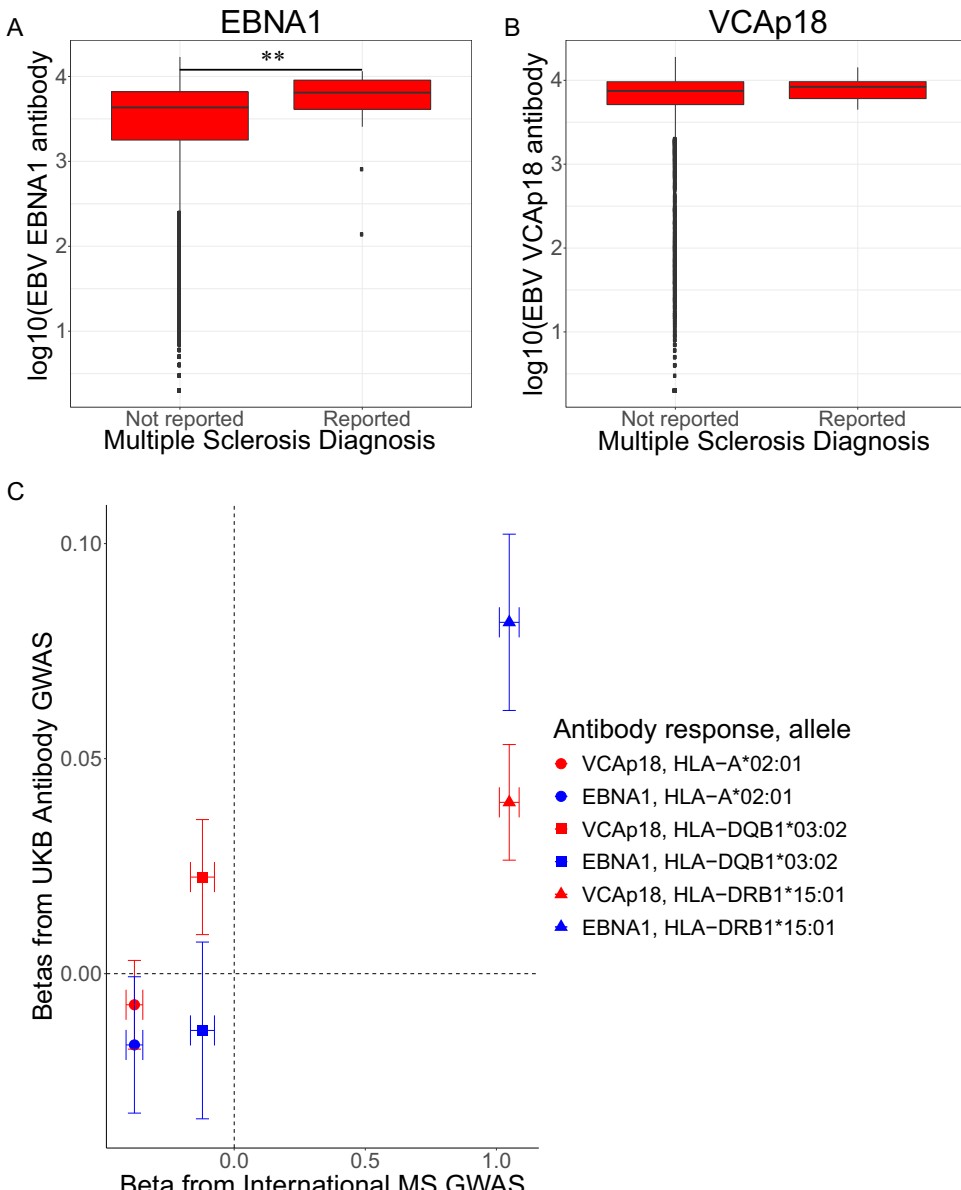

**Fig. 3 EBV EBNA1 and VCAp18 antibody responses and associations with multiple sclerosis.** $Log_{10}$ transformed antibody levels against EBNA1 (**A**) and VCAp18 (**B**) antigens were compared between groups in the UKB Multiplex Serology subset with or without self-reported diagnoses of multiple sclerosis (MS; 34 cases). Box plot centre line, median; box limits, upper and lower quartiles; whiskers, 1.5x interquartile range. **$P = 6.1\times10^{-3}$ using linear regression. **C** The beta coefficients and 95% confidence intervals from the GWAS analyses of quantitative antibody responses against EBNA1 (blue) and VCAp18 (red) in the 9611 biologically independent individuals from the UKB subset (y-axis) were compared to the coefficients and 95% confidence intervals from the largest available case-control GWAS of MS (17,610 cases and 30,129 controls; x-axis) for HLA alleles recognised to be associated with MS risk. The points representing beta estimates are shaped by imputed HLA allele.

compared to the alleles associated with antibody responses against the EBV antigens EBNA1 and VCAp18 in our UKB subset (Fig. 3C). To measure the significance of association between EBV antigen response traits and MS risk more formally we calculated the genetic correlation between quantitative VCAp18 and EBNA1 antibody response associations in UKB against MS case-control status using the same MS HLA GWAS data. We found statistically significant evidence of a positive genetic correlation between MS risk and VCAp18 antibody response using variants across the MHC region alone ($r_G = 0.30$, $P = 0.01$) and a similar level of correlation using variants across the entire genome ($r_G = 0.21$, $P = 0.02$). We did not observe genetic correlation between EBNA1 antibody levels and MS ($r_G = 0.09$, $P = 0.41$). Furthermore using data on coeliac disease

(1,468 cases and 10,000 randomly selected controls) within UKB as a negative control trait where there is known to be a strong HLA association deemed unlikely to have an infectious driver, we found no evidence of genetic correlation between VCAp18 antibody responses and coeliac disease when tested across the genome or across the MHC ($r_G = -0.09$, $P = 0.49$).

## Discussion

Seven viruses and one bacterial species are established causes of cancer. These and many other species have been hypothesised to contribute to a range of NCDs including cardiovascular and inflammatory conditions[1,21–23]. Confirming and understanding the relationships underlying such associations could have

significant implications for public health and facilitate the discovery of novel therapeutics. Here we present data from a Multiplex Serology platform applied to a subset of a very large prospective cohort study. Our findings replicate many established relationships, highlighting the validity of our data, and through these results alongside newly recognised associations we demonstrate the potential of the dataset for addressing multiple questions relating to the epidemiology of infectious agents and biology underlying subsequent disease.

Our seroprevalence estimates were consistent with previously published estimates in the UK and Europe. Some of our seroprevalence estimates were on the higher limit of expectations. For example, the *Ct* seroprevalence in women was approximately 25% in our study, and thus significantly higher than in other studies using the microimmunofluorescence (MIF) assay or major outer membrane protein (MOMP) peptide ELISA[24,25]. However, our seroprevalence estimate was consistent with recent data from England based on the same antigen (pGP3) in ELISA assays[26]. This highly immunogenic antigen is believed to be the most species-specific marker of *Ct* that may provide a more accurate estimate of *Ct* exposure. We have previously demonstrated high agreement of our Multiplex Serology pGP3 assay and the published pGP3 ELISA methods[25].

Our observations of demographic associations with infectious agent seroprevalence, such as those between LSP and sexually transmitted infections like *Ct* or HPV-16 (also reported separately[27]) further enhance the reliability of the seroprevalence estimates derived from our Multiplex Serology panel. Although we did not observe similarly significant associations between LSP and other infections known to be sexually transmitted (HBV, HCV, HIV-1 and HTLV-1), these infections are rare and the existing analysis may be underpowered to detect such associations. Instead, the comprehensive data availability from UKB participants including sensitive aspects of lifestyle meant that we could observe alternative relationships with sexual behaviour reports such as sameSI with HBV, HIV-1 and *Ct* serostatus, thus increasing confidence even in those infections that are rare within UKB.

Many of the demographic associations we observed, such as those with sex, age or deprivation status, have been reported previously but are poorly understood; and such associations may occur as a result of confounding. UKB offers the opportunity to clarify such complex inter-relationships. For example, we were able to demonstrate clear associations with age and CMV[28], HSV-1[29] and *Tg*[30] that are likely to be independent and not confounded by other reported risk factors and we show here the significant utility of follow up samples from individuals within UKB to understand these associations further. Such infectious agents have repeatedly been shown to be transmitted through behaviours in adulthood including close physical and sexual contact, and animal exposure or diet as in the case of *Tg*. Our longitudinal data analysis supports such exposures occurring throughout adulthood and particularly during ongoing UKB follow up for CMV and *Tg* but not for HSV-1 where the relationship is likely to be more complex and our observations may suggest some bias in re-participation of participants and requires further investigation. Our observed pattern of increasing VZV antibody levels over increasing age is in keeping with our understanding of reactivation of the virus over time[14] although we have not tested for reports of shingles (that would be best ascertained through General Practice records) in these individuals that may confirm this speculation. The associations of CMV, HBV and *Tg* we observed with ethnicity were particularly striking. Although these associations remained highly significant after adjustment for age, sex, LSP and TDI, it is likely that these relationships occur as a combination of environmental and genetic factors in addition to the increased likelihood of infectious agent exposure in other countries where participants of alternative ethnicities may have lived when

younger. In contrast, although we observed significant associations between multiple infectious agents and deprivation, these associations were no longer significant after adjustment for age, ethnicity and LSP. Such findings could have significant implications for public health interventions for efforts to improve health equality across strata of society.

The replication of a series of genetic associations with both seropositivity and quantitative antibody responses further increases confidence in the validity of the dataset. Furthermore, the discovery of a signal of association with antibody responses against EBV VCAp18 highlights the potential of this dataset for novel discovery of genetic variant associations. Associations of pathogen-specific antibodies with the MHC locus are mechanistically highly plausible as variation in *HLA* genes may modulate an individual's immune response to an infection influencing susceptibility for and the ability to clear infections, and the magnitude of antibody response[31].

Such genetic associations offer opportunities to improve our understanding of biological mechanisms when interpreted alone. In addition, they can also be used to understand relationships between infections and other NCDs through testing for the presence of shared heritability. This is particularly relevant given the current availability of infectious agent data for only a subset of the existing UKB. Although we had reasonable power within the UKB subset to observe the well-recognised, causative association between HPV-16 seroprevalence and CIN and cervical cancer[32], testing for associations with small to moderate effect sizes is difficult owing to the limited sample size. Nevertheless, when testing a hypothetical association that has long been speculated to be causal, we did still observe differences in the magnitude and seroprevalence of EBV antibody responses in patients with self-reported MS compared to those without MS in directions consistent with previous reports[8]. By comparing genetic association signals derived from the antibody response in the UKB subset against signals generated from an independently recruited case-control analysis of MS, we observed a significant degree of genetic correlation between EBV VCAp18 antibody responses and MS risk. Our estimation with an $r_G$ between 0.2 and 0.3 both across the MHC and the entire genome is in line with previous reports and has been interpreted as evidence in favour of a causal relationship between EBV and MS[33,34]. However, since the majority of this correlation will be driven by the MHC region well known to demonstrate significant levels of pleiotropy (i.e. a single genetic variant may be associated with multiple traits) it is not possible to definitively exclude other confounders that may better explain this relationship. These confounders may indeed include other infectious agents with similar associations across the MHC. Reassuringly we observed no association with coeliac disease that was included as a negative control.

Although we have demonstrated a range of potential utilities for the data generated from the Multiplex Serology panel, it is important to recognise the limitations of our approach. Multiplex Serology is designed to detect IgG antibodies reflecting cumulative (i.e. past or present) exposure, the most relevant information for infectious disease epidemiology. As such, our measurements may not fully correspond to results generated with assays used in clinical diagnosis that may take additional information into account, such as other antibody isotypes, or presence of the pathogen's nucleic acids. Moreover, the multiplex nature of the assay does not allow for the optimisation of assay conditions for each and every infectious agent as demonstrated by the variation in validation statistics (e.g. as seen for *Tg* or HHV-6) nor every potential variation at the human level (such as differing ethnic backgrounds of individuals). Assay precision in this study was good when reporting combined results of seropositives (with low CVs) and seronegatives (with higher CVs); we expect this to

further improve with more automated workflows. Nevertheless, even for *Tg* where we observed lower sensitivity at the used serum dilution, and differential performance on magnetic versus non-magnetic beads, we were still able to observe robust epidemiological associations consistent with the literature, reinforcing the high quality of the data for epidemiological analyses. Based on the described limitations for specific infectious agents including *Tg* and HHV-6 where further validation may be required, caution should be applied in interpreting the results from any analysis involving *Tg* and HHV-6 antigens in the future. We also acknowledge that our findings in non-White ethnicities require replication in larger studies covering a sufficient spread of different ethnicities, as the UK Biobank population is almost exclusively composed of White individuals and furthermore is not fully representative of the UK population as a whole[35]. Irrespective of all of the above, the availability of additional biomaterials (such as saliva and buffy coats) from baseline and follow-up sampling in the UKB participants offers significant future opportunities. For example, it may be possible to undertake complementary assays (such as nucleotide-based detection systems) to add data about acute infections in individuals identified as seropositive for either prior exposure or carriage using data from the Multiplex Serology panel.

We show that our Multiplex Serology panel offers an attractive validated approach to identify both, previously recognised and unrecognised associations between infectious agents and a range of demographic factors and genetic variants that may influence the risk of chronic disease. In time, with further cohort maturation and investigation by the scientific community, these data are likely to have significant impact on our understanding of the burden of infectious agents and their sequelae, which will in turn inform future public health strategies.

## Methods

**Infectious agent selection**. A UK Biobank Infectious Disease Working Group was established to provide a consensus list of infectious agents deemed to be of significant importance to public health (Supplementary Methods), and to determine the most appropriate methodology for their measurement in a large-scale prospective study (Supplementary Table 1). A discussion of the rationale of the selection of the final Multiplex Serology panel (Supplementary Table 2) is discussed in the Supplementary Methods.

**UK Biobank study population**. The UKB recruitment and ethical approval process has been described previously[18]. Briefly, half a million men and women aged 40-69 years attended one of 22 UKB assessment centres located throughout England, Scotland and Wales between 2006 and 2010. All participants completed a touchscreen questionnaire, verbal interview and had a range of physical measurements and blood, urine and saliva samples taken for long-term storage. A subset of 20,000 individuals attended a repeat assessment between 2012 and 2013. UK Biobank has approval from the North West Multi-centre Research Ethics Committee (UK) and informed consent was obtained from all participants.

For this study, serum samples from 9,695 UKB participants were selected at random and assayed using the final Multiplex Serology panel. For 277 (2.9%) of these participants, an additional sample from the repeat assessment was assayed to assess stability of seroconversion over a 4–5 year period (Supplementary Methods).

**Multiplex serology**. Multiplex Serology was developed using the principles of an enzyme linked immunosorbent assay (ELISA) that measures levels of serum immunoglobulin allowing for high-throughput serological measurements by presenting antigens to serum immunoglobulins on beads[15]. In brief, Multiplex Serology is a bead-based glutathione S-transferase (GST) capture assay incorporating glutathione-casein coated fluorescence-labelled polystyrene beads and pathogen-specific GST-X-tag fusion proteins as antigens[15,19]. Full length or fragment viral, bacterial or parasitic antigens were expressed in *E. coli* fused to an N-terminal GST domain and a C-terminal peptide tag for detection of full-length expression and stored as cleared lysate[25,36–40]. Each antigen was loaded onto one distinct bead set via in situ affinity purification of GST fusion proteins on glutathione-derivatised beads. Subsequently, the bead sets (i.e. antigens) were mixed and simultaneously presented to primary serum antibodies (at serum dilution 1:1000). Formed immunocomplexes were detected using a biotinylated goat-α-human IgG secondary antibody and quantified in a Luminex 200 flow cytometer via streptavidin-R-phycoerythrin as reporter dye. Per bead set, at least 100 beads were measured

and Median Fluorescence Intensities (MFI) were calculated. We adapted the originally reported methodology[15] by transferring the platform to magnetic beads to allow for enhanced scalability. The procedure is described in more detail in the Supplementary Methods. For each infectious agent tested, we measured antibody responses for up to six antigens. Where two or more antigens were used, we defined algorithms to combine the responses against multiple antigens and designate overall seropositivity or seronegativity against individual infectious agents.

The workflow for validation of Multiplex Serology including validation of assays for the individual infectious agents and usage of magnetic instead of non-magnetic beads was in line with the criteria outlined in the STARD guidelines[41] (Supplementary Fig. 1). Individual infectious agent immunoassays were validated by comparing seropositivity and seronegativity estimates from the Multiplex Serology platform against gold standard assays using collections of pseudonymised, samples available as reference panels[25,37–40]. The description and validation of the assays for the individual infectious agents described here have been detailed elsewhere (Supplementary Methods and Supplementary Table 2)[15,25,37–40,42,43]. These panels were also used to compare the performance of the agent-specific Multiplex Serology assays conducted once on non-magnetic and once on magnetic beads for the final Multiplex Serology panel (Supplementary Figs. 2 and 3).

**Statistical analysis**. Statistical measures (sensitivity, specificity, Cohen's *kappa*) for Multiplex Serology assays using reference panel samples[25,37–40] including 95% confidence intervals (CI) were calculated using SAS 9.4.

During Multiplex Serology validation (Supplementary Fig. 1) we defined cut-off values distinguishing positive versus negative antibody responses against individual antigens based on previously published analyses[25,37–40,44–46]. These antigen-specific seropositivity results were subsequently combined using published algorithms to define overall seropositivity for infectious agents (as discussed in more detail on the UK Biobank Data Showcase: http://biobank.ctsu.ox.ac.uk/crystal/label.cgi?id=51428).

The seroprevalence estimates for each infectious agent were compared to those expected for European populations based on previously published data, determined through a PubMed search using the terms 'seroprevalence', 'prevalence', 'IgG' or 'antibody' in addition to the abbreviated or full name of the agent (Supplementary Methods). Studies from European populations were included rather than simply focussing on UK studies to ensure the availability of the maximal number of equivalent studies available for comparison with our tested cohort.

Analyses were performed that tested associations between seropositivity (for individual antigens) or overall infectious agent seropositivity (i.e. seroprevalence) as well as $log_{10}$ transformed quantitative MFI antibody responses (i.e., seroreactivity) with environmental, demographic and genetic exposures and disease outcomes. The variables used were defined based on existing knowledge of such associations and are described in the Supplementary Methods. The variables included sex, age (by decade), ethnicity, Townsend deprivation index (TDI) quintiles, lifetime number of sexual partners (LSP in categories of '0', '1', '2-4', '5-10', 'Greater than 10') and self-report of ever same-sex intercourse (sameSI) using a univariate regression model tested against the null (outcome ~ 1) using the likelihood ratio test implemented in the lmtest package (R version 3.5.1). A chi-squared test for trend was used to test for linear upward or downward trends of seroprevalence in the ordered quintiles of TDI. For those variables demonstrating any evidence of association ($P < 0.01$), multivariable logistic regression was used to test for association whilst adjusting for other covariates with or without stratification by sex. Similarly, associations of seropositivity with disease prevalence was performed using either univariate logistic regression or, where regression models could converge with more than 1 individual in each strata assessed, multivariable regression accounting for relevant covariates. Disease outcomes included cervical cancer, multiple sclerosis (MS) and coeliac disease, selected owing to their known (cervical cancer with HPV-16), potential (MS with EBV and CMV exposure) and unlikely (coeliac) association with infection exposure. Disease outcomes were obtained through either self-report, or cancer registry data (details of which are provided in the Supplementary Methods). Antibody levels against multiple infectious agents measured at the baseline and repeat sampling timepoints were compared using a linear mixed model including sex, LSP and TDI quintile as fixed effect covariates and individual IDs as random effect covariates to model the times and ages between sampling for each individual.

**Genetic analysis**. Human genetic variation has been described to influence magnitude of the antibody response against infectious agents that may influence susceptibility to future disease. We used the wealth of data including genetic variation in UK Biobank to replicate previously reported associations between genetic variants and antibody levels and also to assess for genetic correlation between antibody and disease traits. Details of the generation of the genotype data in UKB have been described elsewhere[47] and further details are provided in the Supplementary Methods. To compare association statistics generated from our antibody work against genetic association signals for multiple sclerosis we used available data on genotype variants across the MHC and case-control phenotypes from the International Multiple Sclerosis dataset[20]. HLA alleles were imputed into both the UKB data and the International Multiple Sclerosis datasets using genotype data available from across the extended MHC region only (chromosome 6 base pair positions 25,500,000-34,000,000 in genome build 37) using the SNP2HLA (v1.0.2)

algorithm. Subsequent association analyses were performed in PLINK 1.9 using logistic regression including sex and age as fixed effect covariates, then statistics from individual countries in the International Multiple Sclerosis were meta-analysed using a fixed effects model in METASOFT (v2.0.1). The analyses to test association between genetic variants and antibody traits were performed using all individuals in the UKB subsample using either the GCTA software (version 1.26.0) for directly genotyped variants or BOLT-LMM (version 2.3.1) as a directly equivalent software for imputed data. Both of these algorithms use a linear mixed model to control for cryptic relatedness and to allow inclusion of all individuals irrespective of ethnic origin through the calculation of a genetic relatedness matrix (GRM) incorporated as a random effect covariate. Similarly, a similar approach was used for the methods to test for genetic correlation using bivariate GREML analyses using only directly genotyped variants from European populations. Age and sex were included as fixed effect covariates in all models.

**Reporting summary**. Further information on research design is available in the Nature Research Reporting Summary linked to this article.

## Data availability

All data are available through the UKB Access Management System (https://bbams.ndph.ox.ac.uk/ams/) as described in the following link: https://biobank.ndph.ox.ac.uk/showcase/label.cgi?id=51428. A full list of data fields and identifiers are available in the Supplementary Materials. The antibody data are available in Category 1307. The International Multiple Sclerosis dataset was available through the authors of the publication[20].

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

## Acknowledgements

We are grateful to the researchers and participants in UK Biobank for allowing the collection and use of this data under project code 43920.

## Author contributions

A.J.M., N.B., N.A., M.H., S.S., A.V.S.H. and T.W. conceptualised and designed the methods for the research project; A.J.M., N.B., T.J.L., A.Y.C. and A.C. performed the analyses; A.J.M., N.B., N.A., T.J.L., A.Y.C., R.A., S.S., M.H., G.M., R.C. and T.W. were responsible for generation of resources and curation of data; A.J.M., N.A., G.M., R.C., A.V.S.H. and T.W. were responsible for funding acquisition and supervision, A.J.M., N.B., N.A., T.J.L. and T.W. were responsible for writing the manuscript and all co-authors reviewed the manuscript.

## Funding

AJM was supported by a Wellcome Trust Fellowship with reference 106289/Z/14/Z and by an Oxford University Clinical Academic Graduate School Transitional Fellowship and by the National Institute of Health Research. The research was supported by the Wellcome Trust Core Award Grant Number 203141/Z/16/Z with funding from the NIHR Oxford BRC. The work of T.W. was supported by the Dieter Morszeck Foundation. The views expressed are those of the author(s) and not necessarily those of the NHS, the NIHR or the Department of Health.

## Competing interests

The authors declare no competing interests.

## Additional information

**UKB Infection Advisory Board**

Adrian V. S. Hill[1,7], Allison Aiello[8], Charles Bangham[9], Ray Borrow[10], Judy Breuer[11], Tim Brooks[12], Silvia Franceschi[13], Effrossyni Gkrania-Klotsas[14], Brian Greenwood[15], Paul Griffiths[16], Edward Guy[17], Katie Jeffery[18], Dominic Kelly[19], Paul Klenerman[20], Fiona van der Klis[21], Julian Knight[1], Andrew McMichael[22], Vivek Naranbhai[1], Richard Pebody[23], Tim Peto[24], Andrew J. Pollard[19], Thomas Schulz[25], Kate Soldan[26], Graham Taylor[27], Greg Towers[28], Massimo Tommasino[29], Robin Weiss[11], Denise Whitby[30], Chris Wild[29] & David Wyllie[7]

[8]University of North Carolina at Chapel Hill, Chapel Hill, NC, USA. [9]Department of Medicine, Imperial College London, London, UK. [10]Public Health England, Manchester, UK. [11]Division of Infection and Immunity, University College London, London, UK. [12]Special Pathogens Reference Unit, Public Health England, England, UK. [13]Centro di Riferimento Oncologico, Aviano, Italy. [14]University of Cambridge, Cambridge, UK. [15]London School of Hygiene and Tropical Medicine, London, UK. [16]Royal Free Hospital, London, UK. [17]Public Health Wales, Wales, UK. [18]John Radcliffe Hospital, Oxford, UK. [19]Oxford Vaccine Group, University of Oxford, Oxford, UK. [20]Medawar Building, University of Oxford, Oxford, UK. [21]National Institute for Public Health and Environment, Bilthoven, Netherlands. [22]Nuffield Department of Medicine, University of Oxford, Oxford, UK. [23]Immunisation and Countermeasures Division, National Infections Service, Public Health England, England, UK. [24]Experimental Medicine Division, University of Oxford, Oxford, UK. [25]University of Hannover, Hannover, Germany. [26]Public Health England, England, UK. [27]Imperial College London, London, UK. [28]University College London, London, UK. [29]International Agency for Research on Cancer, Lyon, France. [30]AIDS and Cancer Virus Program, Frederick National Laboratory for Cancer Research, Frederick, MD, USA.

