## [Peer Review File · Nature Communications]

Identification of host-pathogen-disease relationships using a scalable Multiplex Serology platform in UK BiobankReviewers' Comments:

Reviewer #1:

Remarks to the Author:

Nature Comms review July 2020

This a really interesting study showcasing an important use of biobank in the UK. I found some of the paper a little hard to follow and there needs to be improved description of the methods to actually describe what was done. Similarly the text of the results section requires some revision to clearly summarise the results are in the text. In addition, the introduction is too brief.

Comments in detail:

Line 54: Please describe in more details how the lifetime exposure as measured by a cross-sectional IgG can be useful for determining temporality of relationship.

Similarly, in the last section of the introduction could the authors please be clear whether that was part of the current study, or whether that will be included in future studies (ie were the associations assessed temporal or cross-sectional?)

Line 72. Please describe here whether this group was tasked with assessing the important infectious diseases that have links with non-infectious diseases, or infectious diseases in general? If infectious diseases I'm surprised the vaccine preventable and flu etc did not come out. I see from the supplement that these were excluded based on perceived difficulties of assessing within this study design- could this please be made clearer here that this was also part of the assessment for whether pathogens are included in this analysis.

Line 151: Please describe why European populations were chosen and not UK.

"Genetic analysis section". It wasn't quite clear to me how the genetic analysis fit in with the rest of the analysis. Perhaps a further sentence describing how this is analysed in relation to the serological data is necessary in this section. (Some of this is in the results, but it needs to be here)

Line 204. Please provide more detail on the estimates from this study being in line with previous estimates. I think this deserves a table in the main manuscript.

This first section of the results contains sentences telling us where the results are found rather than describing the results. It would be more helpful to have summarising sentences and then (Table 1) in brackets. Eg tell us whether what are the results of the comparison between the selected samples and the main cohort demographics rather than which just that this is shown in table 1. The same for table 2.

Line 209: differences between men and women- needed in brackets where this is shown (Table 2?)

Line 227: Spell out LSP here for the general reader

Were all significant relationships found described in the results? Please make this clear if so and please report other significant if not shown. As the methods is fairly general on what is explored the reader needs to be sure that they are not just being given the explored relationships which gave the results expected and others are not described.

Line 242: This is what needs to be in the methods, and it needs to give some information on what these explored relationships are and how many relationships were tested.

Have all assessments of statistical significance been adjusted for multiple testing. This is particularly important for the genetic associations.

Line 265-: The analysis reported in this section also needs to be described in the methods. I found this section a little hard to follow so suggest some rephrasing in addition to further description of this analysis in the methods.

Line 283- 287: This felt like it would be more helpful in the introduction to provide background and motivation for the study.

Line 293: This is the first time seroprevalence from UK is mentioned. AS above more detail on this in the results this would be very helpful.

Line 361- 365: I'm not quite sure the point here. Suggest rephrasing for clarity.

Reviewer #2:

Remarks to the Author:

Summary: The authors describe the application of a multiplex serology platform capable of quantifying 45 antigens from 20 infectious agents in a pilot set of individuals participating in the UK Biobank effort in order to evaluate associations between infectious diseases exposure and NCDs. They observe differences in the seroprevalence between men and women for 11 infectious diseases, which remained significant after adjustment for relevant covariates.. The authors also identified a statistically significant association between TDI and Hp, after similar adjustments). The authors replicated 3 associations in the class II region of the MHC locus (rs9269910, rs9269268, rs6927022) between genetic variants and antibody responses against JCV, MCV, and EBV EBNA-1, respectively, and were able to identify a novel MHC association (with antibody response to EBV viral capsid antigen (VCAp18). An association was also identified between EBV infection (all four EBV antigens, or VCAp18 alone) and MS providing support for the suspected impact of EBV on MS disease. Overall, the validation of the multiplex serology platform will be of tremendous value when applied to the larger biobank and including additional sampling timepoints. Overall this is a well-conceived and well-run study of general interest. I do have a few questions I would like to see the authors address however.

Major comments

- 1) Why were only 11 reference sera available out of the 20 infectious agents? Were the additional assays validated previously on known control samples?
- 2) More discussion on the limitations of the multiplex data for Tg in the body of the paper would be of benefit to the reader. In supplemental information it is discussed that there are "significant discrepancies in estimated metrics between monoplex and multiplex" with explanations "insufficient volumes remaining for 12% of reference sera, i.e. different numbers of sera tested monoplex versus multiplex validation" and lower sensitivity due to the increased dilution for UKB multiplex panel. Furthermore, when comparing the non-magnetic and magnetic beads the ICC was only 0.48.
- 3) The assay CV ranges are high leading to questions about the reliability of the data. Supplementary Tables 4 and 5. Was any modeling done to determine how this variability would impact the ability to detect evidence of infection, particularly as it relates to seroconversion and reversion in longitudinal sampling.
- 4) I'm a bit confused on precisely how the genetic analyses were run. The methods describe 3 approaches including 1) logistic regression plus meta-analysis across countries (?) using plink 2) bivariate GREML using GCTA (presumably for comparison of genetic correlation) and 3) linear mixed models using BOLT-LMM. It should be clarified which method was used for each result presented why.
- 5) Also on the genetic analysis, the authors present data for SNPs within the HLA and antibody responses to JCV, MCV, and EBV. Since classical allele imputation was performed it would be interesting to also list top classical alleles. As well, given the strong diversity in HLA variation across ancestries it would also be interesting to see how the SNPs and alleles behave stratified by ancestry, power permitting.
- 6) Finally, I think that the access to multiple timepoints is a strength of this paper and the larger study. I think more discussion of this analysis in the main text is warranted.

Minor comments:

- 1) The y axis for Figure 1 B is incorrectly labelled "Seroprevalence in Females (%)"
- 2) In general, the figures are of somewhat poor quality and sections of the text could be edited to

improve readability. For example, the first results section could include descriptive text rather than just pointing the reader to the Table.

Reviewer #3:

Remarks to the Author:

Comments to Manuscript

"Identification of host-pathogen-disease relationships using a scalable Multiplex Serology platform in UK Biobank"

submitted by Mentzer et al

Decision: Major Revision

Overview: The manuscript summarises the initial results of a serology-screening project on the UK Biobank. This is the first of likely multiple papers to arise from the project and aims to prove that the designed serological assay works and can be used to screen the entire biobank or other patient samples to determine serological status. However, in its current form, the manuscript mainly describes basic epidemiological data and gives some limited conclusions about what the dataset could potentially be used for. From my perspective, the most important thing about this piece of work is the multiplex assay, as it has resulted in a method that can be used to screen an infinite number of samples for an impressive number of infectious diseases. Throughout the manuscript, the authors play down this aspect of the work, instead choosing to focus on the epidemiology of their results, and to comment on some issues that can occur with multiplex serology screening. As it is, the data in the manuscript does not provide any novel findings. It confirms data that was already publically available regarding the seroprevalence of the selected diseases within the UK and EU. It is fair to say that the authors use this finding as a validation aspect. While it does nicely show what information could be gained from using the UK Biobank as a resource, this is of limited broad interest and may not suffice publication in this journal. While the multiplex assay should be a major focus of this paper, it also does not meet the requirements for novelty as most (maybe all) of these markers have been previously published elsewhere. Finally, the manuscript currently contains no clear methodological details on how the assay was performed. To my understanding, this is the first time that this group has used magnetic beads, thus the protocol should be described in more detail. Furthermore, the authors should give more attention to the technical and clinical validation. As it would not currently be possible to perform the repeat the assay based upon the details currently submitted, this makes it impossible to validate their results. As a result, the manuscript needs major revision.

Major Comments:

1. Serology assay methods. The full protocol for the assay is never described in either the manuscript or the supplementary materials. This currently makes the assay impossible to reproduce which in turn could add questions to quality of the results. The authors need to provide a detailed protocol for how exactly the assay was performed including full details on coupling, measuring, QC and running the assay with the biobank serum. How do they decide a plate run to be successfully processed or failed? Levy-Jennings-Plots / Westgard rules may be applied. 3 QC samples seems to me a very low number to control 45 antigens along a large screening. Are these samples specifically designed to meet the necessary criteria?

Do the authors control the addition of a sample and detection system to a well? For readers not familiar with this kind of assays and large screening applications, it is difficult to understand how the QC samples are processed and blind-spiked samples are investigated. The methods must be substantially expanded and the QC thoroughly explained in the manuscript, before the accuracy of their results can be validated.

2. While the paper is well-written, it is currently focused towards or too strongly towards the wrong topic. The current focus of this paper is towards epidemiological observations that can be determined

based on the serology of the sample subset. These observations in the form of seroprevalences seem to be similar to what other authors have previously published. Therefore this focus not only substantially reduces the novelty, but also the potential impact of the work, as it can be argued that ~9000 samples is insufficient to make broad epidemiological arguments. Instead, I encourage the authors to rewrite the manuscript with a far greater focus towards the serology multiplex assay aspect. While it does not need to be 50:50 with the epidemiology, it does need to be properly addressed and focused upon, as this assay and the potential now to do high-throughput screening on an unlimited number of samples for 20 infectious agents, is the most important aspect of this work.

3. Serology assay methods part 2. The reasons behind the selection of each disease is currently unclear. In the manuscript, the authors state a working group was established to provide a consensus list of infectious agents deemed to be important to public health and that this is listed in the Supplementary Materials. When viewing the supplementary materials, supplementary table 1 only states whether a disease was carried forward or not to panel. There is no explanation or reference anywhere to why any of the diseases would be selected in the first place, other than the panel appears to have selected them. There needs to be some additional columns added to Supp. Table 1 and the Supp. Materials in general explaining the rationale behind the choice of every disease.

4. Gold standard assays – the authors state that these were used to validate the individual infectious agent immunoassays readers have to collect the information on the assays used from previous publications are, nor could I find any mention of them being performed or how they may have been performed in the supplementary methods. The authors need to include some details (even if it's just the name of the test that they did), even if the results of these assays have been previously described in previous publications.

In addition, they have to describe whether they have used appropriate sample sets for the clinical validation for the different infective agents with focus on the different ethnicities taken into account in the current manuscript. Currently, it appears that their test is working for European citizens but not necessarily for samples derived from individuals from other continents/regions of the world. This leads to the question, whether the cutoffs are not appropriate for the samples from black individuals. Finally, the found sensitivity and specificity for each infective agent should be provided. Why are the samples shown in two different graphs for CMV and HBV. Being from a "different set" is not a proper reason for different sensitivity /specificity calculations. How would you finally determine the sensitivity /specificity of the assay?

5. Cut off values for each antigen tested are not included within the manuscript or its supplementary materials. How do they differ between batches? How are they adjusted in case of batch differences? As I understood it, a kind of value assignment is made by testing a distinct set on each batch, right? Are cutoff samples used?

I think the paper can gain from the inclusion of follow up samples. However, it is not clear to me whether the found differences in seroprevalence are because of samples being close to the cutoff and are therefore ranked differently due to technical variations?

6. Batch-to-batch variation (supplementary Figure 3)

As this is supplementary, why not show it for all antigens instead of selecting only 4.

Fig.3B slopes are missing. Choosing the same max for the x- and y-axis allows a better estimation of comparability. How comparable are the other days? When not depicting, listing of correlation coefficients might be an alternative. As the sample values look not normally distributed, a Pearson correlation might not be the best choice.

7. Technical validation. All information on technical validation are placed in the supplementary part. For a technical validation, I would at least expect to have the intra- and inter-assay variance tested. The authors used the described screen to collect data for the inter-assay variance but do not mention intra-assay variance. How do both look like in the lower, mid and upper MFI response for the used antigens? Have they tested for the impact of freeze-thaw cycles on the antibody response? Have they

tested for the impact of rheumatoid factors, HAMA, etc.?

8. Figure 1 is split on gender and only features on the diseases examined, yet the text appears to report the average for all samples. Please alter figure 1 or the text as such so that they are the same. Similarly, I would like to see either all diseases represented in the graph or diseases for which there are significant differences (evidence of significant between the different groups is also currently lacking, if no interactions are significant, then please state so in the figure legend) between the various cohorts (age, ethnicity and LSP). Additionally, the text for this figure only reports on higher CMV in Asian compared to White and HBV in Black compared to white individuals. Looking at the data as presented, it seems like black and Asian cohorts have higher seroprevalence than the white cohort for all diseases shown. The authors may wish to mention this in the text as it appears significant for some diseases (e.g. CMV). The authors do not also comment at any point about potential problems with the UKB. It appears to be chronically underrepresentative for all non-white ethnicities as well as for individuals with a higher degree of socio-economic deprivation, as compared to the current UK census, in addition to the stated youngest age of 40. Additionally, Fig 1A and B look pretty much the same. Please check, in Fig1B should be results on males presented not on females.

9. The number of samples listed in Supplementary Page 4 does not tally on more than one occasions. The 29 samples excluded is currently 28 based on their description (1 viscous, 8 pipetting errors, 8 incorrect dilutions and 11 insufficient bead counts). The 10110 serum samples also does not tally, as it is currently 10108 based upon their description.

10. Sample variance seems very high to me between days and plates. 16% plate to plate (max 23%) and 21% day to day (max 26%).

11. High seroprevalences in the black group are astonishing. Is this to be expected? Here it might be necessary to check the cutoffs whether they are appropriately adjusted to this ethnical group. I do understand that it will be difficult to get the respective samples, but with the view on the screening of the UKB samples clinical validation for this ethnical subgroups is recommended.

Minor Comments:

1. Introduction – the current introduction is light or missing on information related to either the diseases that would be examined and multiplex serology generally. I encourage the authors to add a short paragraph explaining both what diseases would be studied along with the rational behind their inclusion. I would also encourage the authors to add a short paragraph explaining why they chose to use multiplex bead assay serology and how it has been used before in similar projects.
2. Supplementary Table 4 and 6 present a different number of antigens. Should be 45 antigens.
3. Supplementary Fig. 4. Missing p101k

Page 3 – national-scale instead of prospective. It is already in progress so it cannot really be prospective.

Page 3 – reference missing for “Moreover, since many infectious agents such as Epstein-Barr virus (EBV), cytomegalovirus (CMV) and human immunodeficiency virus (HIV-1) are known to have modulatory effects on the immune system”

Page 14 – first sentence of discussion needs a reference as it is presented as a statement of fact

Page 11 Supp – figure legend need to read B, D, F and H.

NCOMMS-20-24269A REVIEWER COMMENTS

Reviewer #1 (Remarks to the Author):

Nature Comms review July 2020

1. This a really interesting study showcasing an important use of biobank in the UK. I found some of the paper a little hard to follow and there needs to be improved description of the methods to actually describe what was done. Similarly the text of the results section requires some revision to clearly summarise the results are in the text. In addition, the introduction is too brief.

We thank this reviewer for their positive and encouraging comments. In light of these comments and those from all reviewers, we have substantially altered the manuscript by expanding the Introduction and Results sections, further emphasising the technical aspects and unique utility of the Multiplex Serology platform for scalable assessment in very large epidemiological cohorts, and we have also restructured the Methods to provide a series of descriptions that are hopefully easier to follow. As per Editorial suggestion, we have moved the Methods to the later part of the document. We are grateful for all raised points and feel that making the changes in light of these comments has substantially improved the manuscript.

Comments in detail:

2. Line 54: Please describe in more details how the lifetime exposure as measured by a cross-sectional IgG can be useful for determining temporality of relationship.

As part of the expansion of our Introduction, we now describe the epidemiological benefits of a prospective cohort study where measuring exposure against infections at baseline sample and longitudinal follow up and adjustment for other potential confounders should provide more reliable estimation of risk and causality attributable towards infectious exposures to chronic diseases. We hope this is acceptable to the reviewer. Specifically we provide a detail of rationale in lines 70-76 in the new manuscript:

"Definitive evidence to support or refute these postulated associations are most likely to come from prospective cohort studies. In such studies it is possible to measure potential exposures (be they infectious, lifestyle or inherent to the individual such as genetic) in a sample of the population at baseline recruitment and then follow up individuals until they develop disease to thus define exposure and disease temporality."

And for UKB specifically in lines 110-112:

"The application of such technology in UKB offers the opportunity to test for the risk of prior exposure to infectious agents (through cross-sectional antibody measure), and subsequent incidence of cases of a disease of interest, adjusting for potential confounders that should provide a more reliable estimation of risk and causality. "

3. Similarly, in the last section of the introduction could the authors please be clear whether that was part of the current study, or whether that will be included in future studies (ie were the associations assessed temporal or cross-sectional?)

The majority of the reported analyses in this manuscript are indeed cross-sectional and we have highlighted this in the final paragraph of the Introduction, as proposed (lines 117-122):

“Identification of host-pathogen-disease relationships using a scalable Multiplex Serology platform in UK Biobank”

“Using a randomly selected subset of 9,695 individuals, we demonstrate the high performance characteristics of the platform specifically adapted for high throughput and demonstrate the compatibility with prior single agent studies through confirming expected seroprevalence estimates and reproducing previously reported cross-sectional epidemiological and genetic associations with infectious agent exposure that may have utility in a range of future explorations.”

4. Line 72. Please describe here whether this group was tasked with assessing the important infectious diseases that have links with non-infectious diseases, or infectious diseases in general? If infectious diseases I’m surprised the vaccine preventable and flu etc did not come out. I see from the supplement that these were excluded based on perceived difficulties of assessing within this study design- could this please be made clearer here that this was also part of the assessment for whether pathogens are included in this analysis.

The Expert Working Group were tasked with identifying infectious agents of relevance to UK public health whilst also considering technical factors influencing the selection of assay to use and the antigens to include. We have detailed this aspect further in the Results section (line 142) that is the first section the reader will be exposed to now that we have rearranged the text as per Editorial instruction. A greater detail of specifics relating to the selection of specific infectious agents is now included in the Supplementary Methods where we also highlight the rationale for consideration and exclusion of the vaccine-preventable diseases due to perceived difficulty in collating the number of doses of vaccine administered to individuals that would confound any potential associations.

5. Line 151: Please describe why European populations were chosen and not UK.

The aim of this study was to estimate prior exposure to a large number of infectious agents in the UK Biobank sample and to compare this to published estimates. Unfortunately such published estimates were not universally available for all agents of interest. Thus we decided to expand the scope of studies to encompass European populations as well as the UK, which should serve as a reasonable equivalent proxy for the UK Biobank estimates.

We have added the following sentence to make this clearer in the Methods section (lines 494-497):

“Studies from European populations were included rather than simply focussing on UK studies to ensure the availability of the maximal number of equivalent studies available for comparison with our tested cohort.”

6. “Genetic analysis section”. It wasn’t quite clear to me how the genetic analysis fit in with the rest of the analysis. Perhaps a further sentence describing how this is analysed in relation to the serological data is necessary in this section. (Some of this is in the results, but it needs to be here)

We thank the reviewer for this suggestion. We have substantially revised the Results section and inserted the following sentences at the beginning of the genetic analysis Methods section that we hope makes our analytical approach rationale clearer (lines 524-528):

“Human genetic variation has been described to influence the magnitude of antibody response against infectious agents that may influence susceptibility to future disease. We used the wealth of data including genetic variation in UK Biobank to replicate previously reported associations

“Identification of host-pathogen-disease relationships using a scalable Multiplex Serology platform in UK Biobank”

between genetic variants and antibody levels and also to assess for genetic correlation between antibody and disease traits. “

7. Line 204. Please provide more detail on the estimates from this study being in line with previous estimates. I think this deserves a table in the main manuscript.

The previous estimates were indeed already summarised with references in Table 2 of the main manuscript and have been retained as per the reviewer’s comment.

8. This first section of the results contains sentences telling us where the results are found rather than describing the results. It would be more helpful to have summarising sentences and then (Table 1) in brackets. Eg tell us whether what are the results of the comparison between the selected samples and the main cohort demographics rather than which just that this is shown in table 1. The same for table 2.

We thank the reviewer for this suggestion. We have now presented an overview of the comparisons previously directed to in the Tables to the main text with some specific outlined examples throughout the Results text.

9. Line 209: differences between men and women- needed in brackets where this is shown (Table 2?)

We thank the reviewer for pointing out this omission. We have included reference to Table 2 for this observation.

10. Line 227: Spell out LSP here for the general reader

We would like to highlight that we had already spelt out LSP in the Methods section in statistical analysis. However, with the restructuring of the text for Nature Communications this will now be described later in the text and have made the changes accordingly. We agree that this helps significantly improve the flow of the reading of the manuscript.

11. Were all significant relationships found described in the results? Please make this clear if so and please report other significant if not shown. As the methods is fairly general on what is explored the reader needs to be sure that they are not just being given the explored relationships which gave the results expected and others are not described.

We agree with the reviewer that it is important to highlight to the readers which analyses were done and reported. We have detailed in the Methods and further in the Supplementary Methods how we undertook our analyses aiming to present only those demographic features that were found to be associated with each infectious agent after crude testing. For the demographic analyses, the results from all infectious agents with any significant association following adjustment are now in the main Figure for transparency (also in answer to a point raised by Reviewer 3) and some are presented in the text. Furthermore, the Supplementary Tables list all crude associations and the exploration of associations following adjustment for important covariates to enable a more transparent interpretation. For disease and genetic associations, only *a priori* hypothesised associations were tested and the results of all of these analyses are presented in the relevant results sections.

“Identification of host-pathogen-disease relationships using a scalable Multiplex Serology platform in UK Biobank”

12. Line 242: This is what needs to be in the methods, and it needs to give some information on what these explored relationships are and how many relationships were tested.

We thank the reviewer for this suggestion. We have adapted the text in a way that we hope makes the rationale and steps used in the genetic analyses clearer both in the Methods and the Results sections to address this point as well as points 8. and 11. above.

13. Have all assessments of statistical significance been adjusted for multiple testing. This is particularly important for the genetic associations.

All genetic association tests surpass the statistics required accounting for multiple testing. This is described in the Results section (lines 265-268):

“All these reported associations remain highly significant even after accounting for multiple testing using Bonferroni correction and a stringent GWAS P-value threshold (nominally $P=5 \times 10^{-9}$ adjusted to 1×10^{-9} for five individual GWAS).”

The other demographic and disease associations were undertaken in light of previously published findings so even a nominal significant result ($P \leq 0.05$) with an effect in the anticipated direction would constitute a valid result. Nevertheless, many reported associations remain highly significant even considering the multiple testing undertaken (as demonstrated in the Supplementary Tables where many significant observations have P values less than 1×10^{-3}). We have not formally calculated a threshold for what would be acceptable as we feel the tests are not all independent and a Bonferroni calculation would be overly stringent but feel the full presentation of statistics allows for the reader to make an informed decision alongside our interpretation of the results.

14. Line 265-: The analysis reported in this section also needs to be described in the methods. I found this section a little hard to follow so suggest some rephrasing in addition to further description of this analysis in the methods.

We have significantly amended the text in the Methods section in a way that we hope improves readability and understanding in line with comments from this, and other, reviewers.

15. Line 283- 287: This felt like it would be more helpful in the introduction to provide background and motivation for the study.

We thank the reviewer for this suggestion. We have expanded on the highlighted sentence to provide more of a backdrop to our approach in the Introduction.

16. Line 293: This is the first time seroprevalence from UK is mentioned. AS above more detail on this in the results this would be very helpful.

We have detailed why European study seroprevalence estimates are used in some circumstances in the Results section in order to compare our results to published associations between demographic factors and infectious agent seropositivity where we may not have comparative UK data available (lines 187-189):

“Identification of host-pathogen-disease relationships using a scalable Multiplex Serology platform in UK Biobank”

“...we found our estimates to be within the estimate ranges reported in the literature for equivalent British or, if not available other European or worldwide populations...”

17. Line 361- 365: I'm not quite sure the point here. Suggest rephrasing for clarity.

We have rephrased the sentence to improve clarity as (lines 396-406):

“Although we have demonstrated a range of potential utilities for the data generated from the Multiplex Serology panel, it is important to recognise the limitations of our approach. Multiplex Serology is designed to detect IgG antibodies reflecting cumulative (i.e. past or present) exposure, the most relevant information for infectious disease epidemiology. As such, our measurements may not fully correspond to results generated with assays used in clinical diagnosis that may take additional information into account, such as other antibody isotypes, or presence of the pathogen’s nucleic acids. Moreover, the multiplex nature of the assay does not allow for the optimisation of assay conditions for each and every infectious agent as demonstrated by the variation in validation statistics (e.g. as seen for Tg) nor every potential variation at the human level (such as differing ethnic backgrounds of individuals).”

Reviewer #2 (Remarks to the Author):

Summary: The authors describe the application of a multiplex serology platform capable of quantifying 45 antigens from 20 infectious agents in a pilot set of individuals participating in the UK Biobank effort in order to evaluate associations between infectious diseases exposure and NCDs. They observe differences in the seroprevalence between men and women for 11 infectious diseases, which remained significant after adjustment for relevant covariates.. The authors also identified a statistically significant association between TDI and Hp, after similar adjustments). The authors replicated 3 associations in the class II region of the MHC locus (rs9269910, rs9269268, rs6927022) between genetic variants and antibody responses against JCV, MCV, and EBV EBNA-1, respectively, and were able to identify a novel MHC association (with antibody response to EBV viral capsid antigen (VCAp18). An association was also identified between EBV infection (all four EBV antigens, or VCAp18 alone) and MS providing support for the suspected impact of EBV on MS disease. Overall, the validation of the multiplex serology platform will be of tremendous value when applied to the larger biobank and including additional sampling timepoints. Overall this is a well-conceived and well-run study of general interest. I do have a few questions I would like to see the authors address however.

We thank the reviewer for the positive interpretation of our work.

Major comments

1) Why were only 11 reference sera available out of the 20 infectious agents? Were the additional assays validated previously on known control samples?

We agree with the reviewer’s note that most of the other infectious agents’ assays were already validated in other studies before. For some infectious agents, such as human herpesviruses 6-8,

Multiplex Serology could not be validated sufficiently since reliable serological reference assays are currently lacking. For other infections we believe that the established use of the Multiplex Serology method in a wide range of previous studies made any further validation redundant. We have summarized all references for the assay validations in Supplementary Table 2.

2) More discussion on the limitations of the multiplex data for Tg in the body of the paper would be of benefit to the reader. In supplemental information it is discussed that there are "significant discrepancies in estimated metrics between monoplex and multiplex" with explanations "insufficient volumes remaining for 12% of reference sera, i.e. different numbers of sera tested monoplex versus multiplex validation" and lower sensitivity due to the increased dilution for UKB multiplex panel. Furthermore, when comparing the non-magnetic and magnetic beads the ICC was only 0.48.

We thank the reviewer for pointing out the limited performance of the Tg assay and highlighting the need for further discussion in the manuscript. Thus, we added two sentences to the limitation section in the Discussion where Tg assay performance was already mentioned suggesting caution with the use of serological measurements against Tg in future use within the UKB cohort (lines 411-413):

"Based on the described limitations for Tg, caution should be applied in interpreting the results from any analysis involving Tg antigens in the future."

3) The assay CV ranges are high leading to questions about the reliability of the data. Supplementary Tables 4 and 5. Was any modeling done to determine how this variability would impact the ability to detect evidence of infection, particularly as it relates to seroconversion and reversion in longitudinal sampling.

We wish to respectfully disagree with this assessment. Supplementary Table 4 shows coefficients of variation (CV) for 107 blind spiked duplicates, once for all samples, and once among seropositives only. Even among all samples, the median CV across all antigens is 17% which is generally considered acceptable (i.e., <20%) for biological assays (e.g., Bower KM. Statistical Assessments of Bioassay Validation Acceptance Criteria. *BioProcess Int.* 16(8) 2018). More importantly, we do not feel that it is particularly meaningful to calculate CVs among seronegatives, as 1) they are negative by definition, so any quantitative analysis of antibody levels is hard to interpret, especially below the lower limit of quantitation (approx. 30 MFI at 1:1000 serum dilution), and 2) the CV tends to be inflated for small numbers. Among the seropositives, the median CV is 3.5% which is typically considered excellent, with a maximum of 12%. Repeatability and reproducibility of multiplex serology have been assessed numerous times in similar settings, i.e. using blind interspersed duplicates in international cohort consortia, and are typically described as excellent, with intra-class correlation coefficients (ICC) and Pearson correlation coefficients both approaching 1.0 (e.g. Kreimer *et al.* *Ann Oncol* 2019, Butt *et al.* *Gastroenterology* 2018). Thus, we do not expect major impacts of measurement variability on detecting evidence of infection.

Of course, even the smallest assay variation would inevitably lead to some discordant reproducibility results, or fluctuations (i.e., seroconversion and -reversion) in longitudinal sampling, when the quantitative MFI values are dichotomized using cut-offs. We have critically discussed this in the Supplementary Methods to Supplementary Table 5 ("... these mismatches are likely a result of the CV of the assay and the remaining 10-30% of discordant samples are likely to represent true seroconverters or reverters").

“Identification of host-pathogen-disease relationships using a scalable Multiplex Serology platform in UK Biobank”

In order to address the reviewer’s concern, we have added explanatory text to the Supplementary Methods, and summarized the main results from Supplementary Tables 4 and 5 in the main text. Of note, we have critically discussed the performance of our *Toxoplasma gondii* assay in the updated version of the manuscript as per comment 2 above, and have added assay precision in the limitations section of the Discussion.

4) I’m a bit confused on precisely how the genetic analyses were run. The methods describe 3 approaches including 1) logistic regression plus meta-analysis across countries (?) using plink 2) bivariate GREML using GCTA (presumably for comparison of genetic correlation) and 3) linear mixed models using BOLT-LMM. It should be clarified which method was used for each result presented why.

We have elaborated further on the rationale for the use of the varied methods used in the genetics analyses both in the Methods and Results sections that we hope make this more straightforward to follow for both the reviewer and the reader.

5) Also on the genetic analysis, the authors present data for SNPs within the HLA and antibody responses to JCV, MCV, and EBV. Since classical allele imputation was performed it would be interesting to also list top classical alleles. As well, given the strong diversity in HLA variation across ancestries it would also be interesting to see how the SNPs and alleles behave stratified by ancestry, power permitting.

We are grateful for this suggestion. We have now presented the most significantly associated HLA alleles for the tested antibody traits using the imputed allele data in Supplementary Table 14. These consistently show less significant association than with the SNPs that is frequently observed in such analyses. We have undertaken some careful stratification of individuals by ancestry and analysed the SNP associations. However, since the numbers of individuals in non-European ancestral groups are very limited (148 African, 49 Asian) we have not provided the results in the manuscript. For the benefit and interest of the Reviewer we find similar effect estimates for all index associated SNPs in each ancestry for all tested traits. We have not analysed the HLA alleles in a similar way since four-digit coding of alleles are often stratified by ancestry and therefore such data is unlikely to be informative.

6) Finally, I think that the access to multiple timepoints is a strength of this paper and the larger study. I think more discussion of this analysis in the main text is warranted.

We thank the reviewer for this excellent suggestion. We agree that the resampling is a significant benefit to the study. Indeed, based on this observation, we have undertaken a new analysis using this data to explore the relationships between age, or time, and seropositivity to multiple infectious agents in the limited number of individuals with additional visit data available. We have presented these findings in the results of demographic associations (lines 210-226) and used these as exemplar analyses in the Discussion (lines 348-356). We hope the reviewer agrees that despite the small number of samples included these results are interesting and informative for future analyses.

Minor comments:

1) The y axis for Figure 1 B is incorrectly labelled “Seroprevalence in Females (%)”

“Identification of host-pathogen-disease relationships using a scalable Multiplex Serology platform in UK Biobank”

Based on the feedback from multiple reviewers the demographic figures have been redone and this legend axis has been corrected. We apologise for the error in the first draft.

2) In general, the figures are of somewhat poor quality and sections of the text could be edited to improve readability. For example, the first results section could include descriptive text rather than just pointing the reader to the Table.

We have amended much of the text in light of all reviewer’s comments. Furthermore we have updated Figure 1 that hopefully improves the quality for the benefit of the reader.

Reviewer #3 (Remarks to the Author):

Comments to Manuscript

“Identification of host-pathogen-disease relationships using a scalable Multiplex Serology platform in UK Biobank”

submitted by Mentzer et al

Decision: Major Revision

Overview: The manuscript summarises the initial results of a serology-screening project on the UK Biobank. This is the first of likely multiple papers to arise from the project and aims to prove that the designed serological assay works and can be used to screen the entire biobank or other patient samples to determine serological status. However, in its current form, the manuscript mainly describes basic epidemiological data and gives some limited conclusions about what the dataset could potentially be used for. From my perspective, the most important thing about this piece of work is the multiplex assay, as it has resulted in a method that can be used to screen an infinite number of samples for an impressive number of infectious diseases. Throughout the manuscript, the authors play down this aspect of the work, instead choosing to focus on the epidemiology of their results, and to comment on some issues that can occur with multiplex serology screening. As it is, the data in the manuscript does not provide any novel findings. It confirms data that was already publically available regarding the seroprevalence of the selected diseases within the UK and EU. It is fair to say that the authors use this finding as a validation aspect. While it does nicely show what information could be gained from using the UK Biobank as a resource, this is of limited broad interest and may not suffice publication in this journal. While the multiplex assay should be a major focus of this paper, it also does not meet the requirements for novelty as most (maybe all) of these markers have been previously published elsewhere. Finally, the manuscript currently contains no clear methodological details on how the assay was performed. To my understanding, this is the first time that this group has used magnetic beads, thus the protocol should be described in more detail. Furthermore, the authors should give more attention to the technical and clinical validation. As it would not currently be possible to perform the repeat the assay based upon the details currently submitted, this makes it impossible to validate their results. As a result, the manuscript needs major revision.

Major Comments:

1. Serology assay methods. The full protocol for the assay is never described in either the manuscript or the supplementary materials. This currently makes the assay impossible to reproduce which in turn could add questions to quality of the results. The authors need to provide a detailed protocol for how

“Identification of host-pathogen-disease relationships using a scalable Multiplex Serology platform in UK Biobank”

exactly the assay was performed including full details on coupling, measuring, QC and running the assay with the biobank serum. How do they decide a plate run to be successfully processed or failed? Levy-Jennings-Plots / Westgard rules may be applied. 3 QC samples seems to me a very low number to control 45 antigens along a large screening. Are these samples specifically designed to meet the necessary criteria?

We thank the reviewer for pointing out that the assay procedure should be explained in more detail. In light of this observation we added a dedicated section in the Supplementary Methods, and referenced previous publications describing the methodology and expanded on differences / special handling for this study.

In addition, we also described the quality control measures in more detail. Please note that the 3 QC samples were not used to control all 45 antigens across all plates but to detect plate-to-plate variance, i.e. plate handling. The antibody levels of these 3 sera were monitored across all plates per assay day (n=20 plates per assay day; 6 assay days) to detect deviances for single plates which would have suggested improper handling of this plate. All plates passed this quality control check. We included an example plot (Supplementary Figure 5) for one of the 3 standard sera in the Supplementary Methods section.

2. Do the authors control the addition of a sample and detection system to a well? For readers not familiar with this kind of assays and large screening applications, it is difficult to understand how the QC samples are processed and blind-spiked samples are investigated. The methods must be substantially expanded and the QC thoroughly explained in the manuscript, before the accuracy of their results can be validated.

We thank the reviewer for pointing out that the QC section needs further expansion. The addition of sample and assay reagents are monitored by the lab personnel and cross-checked during data processing and quality control checks. We have described these quality control checks in detail in the Supplementary Material (section Multiplex Serology Protocol and UK Biobank Multiplex Serology quality control).

3. While the paper is well-written, it is currently focused towards or too strongly towards the wrong topic. The current focus of this paper is towards epidemiological observations that can be determined based on the serology of the sample subset. These observations in the form of seroprevalences seem to be similar to what other authors have previously published. Therefore this focus not only substantially reduces the novelty, but also the potential impact of the work, as it can be argued that ~9000 samples is insufficient to make broad epidemiological arguments. Instead, I encourage the authors to rewrite the manuscript with a far greater focus towards the serology multiplex assay aspect. While it does not need to be 50:50 with the epidemiology, it does need to be properly addressed and focused upon, as this assay and the potential now to do high-throughput screening on an unlimited number of samples for 20 infectious agents, is the most important aspect of this work.

We thank the reviewer for his/her appreciation of our methodology. We have considerably expanded upon the methods, including protocols, similarities and differences compared to previously published studies, and assay validation. We do however believe that an epidemiological study of over 9,000 individuals is an asset in itself – few studies have ever investigated infectious disease biomarkers at this scale in a prospective cohort. Our epidemiological findings represent both validation of methods and results, and showcase the ability to generate novel findings, all of which have been appreciated by Reviewers 1 and 2 above. We hope with the considerable

“Identification of host-pathogen-disease relationships using a scalable Multiplex Serology platform in UK Biobank”

extension of the methodological aspects in the revised version, the balance between methods and epidemiology (that we personally feel is more of interest to the general reader) is acceptable for the reviewer.

4. Serology assay methods part 2. The reasons behind the selection of each disease is currently unclear. In the manuscript, the authors state a working group was established to provide a consensus list of infectious agents deemed to be important to public health and that this is listed in the Supplementary Materials. When viewing the supplementary materials, supplementary table 1 only states whether a disease was carried forward or not to panel. There is no explanation or reference anywhere to why any of the diseases would be selected in the first place, other than the panel appears to have selected them. There needs to be some additional columns added to Supp. Table 1 and the Supp. Materials in general explaining the rationale behind the choice of every disease.

As stated in the text belonging to Supplementary Table 1 (Supplementary Methods, Chapter 2. “Infectious agents, antigen selection and prioritisation”, p.3), the Working Group were asked to suggest a range of infectious agents that were linked with the development of chronic disease and of likely relevance for UK public health. Supplementary Table 1 lists every pathogen that was brought forward, and we have now added a column with the specific reasons for each pathogen, i.e. the putative disease links of interest. We have also expanded the abovementioned text to explain the general rationale behind pathogen choice in more detail.

4. Gold standard assays – the authors state that these were used to validate the individual infectious agent immunoassays readers have to collect the information on the assays used from previous publications are, nor could I find any mention of them being performed or how they may have been performed in the supplementary methods. The authors need to include some details (even if it’s just the name of the test that they did), even if the results of these assays have been previously described in previous publications.

As per the reviewer’s request, we have now added the gold-standard tests used for assay validation to Supplementary Table 2. The assay sensitivity, specificity, and agreement with the gold-standard assay (kappa) is shown in Supplementary Figure 2.

5. In addition, they have to describe whether they have used appropriate sample sets for the clinical validation for the different infective agents with focus on the different ethnicities taken into account in the current manuscript. Currently, it appears that their test is working for European citizens but not necessarily for samples derived from individuals from other continents/regions of the world. This leads to the question, whether the cutoffs are not appropriate for the samples from black individuals.

Our assays were not validated against gold-standard assays using reference samples obtained from different ancestral groups – we agree with the reviewer this would be a desirable, but highly complex undertaking (e.g., African Americans may be different from Sub-Saharan Africans, Chinese from Japanese etc.). However, to the best of our knowledge, there is little evidence for ancestral differences in the antibody response to infectious agents, beyond immunogenetic differences (esp. HLA) which can be controlled for in the UK Biobank. The only example known to us, even though rather geographical rather than strictly ancestral, is the higher unspecific background observed in sera collected from areas where malaria is endemic, presumably due to the distorted antibody affinity maturation process triggered by *Plasmodium* species. However, this does certainly not apply to non-white populations in the UK, and we believe the UK Biobank population (94% whites)

“Identification of host-pathogen-disease relationships using a scalable Multiplex Serology platform in UK Biobank”

is not appropriate to study this question in detail. To take the reviewer comment into account, we have added this as a limitation to the Discussion section.

6. Finally, the found sensitivity and specificity for each infective agent should be provided. Why are the samples shown in two different graphs for CMV and HBV. Being from a “different set” is not a proper reason for different sensitivity /specificity calculations. How would you finally determine the sensitivity /specificity of the assay?

The sensitivities, specificities and Cohen’s kappa are provided in Supplementary Figure 2.

We thank the reviewer for making us aware that for CMV and HBV the inclusion of two graphs lead to confusion. For both infectious agents, we were able to compare our assay against two other reference assays using different reference serum panels. Thus, slightly different performance of our CMV / HBV assay in comparison with the different reference assays are probably due to the differential characteristics and performance of the reference assays. For both infectious agents, the sensitivities and specificities in comparison to the reference assays was high, and we reported the results of both validation approaches for completeness. We clarified this in the figure legend of Supplementary Figure 2.

5. Cut off values for each antigen tested are not included within the manuscript or its supplementary materials. How do they differ between batches? How are they adjusted in case of batch differences? As I understood it, a kind of value assignment is made by testing a distinct set on each batch, right? Are cutoff samples used?

The same cut-offs were applied to all samples. We used 184 samples which were tested on each assay day of an assay week, and another set of 184 samples which were tested on the first day of assay week 1 and 2 to monitor assay drift and normalize the data if needed. We added a detailed description of the approach in the Supplementary Methods (section Multiplex Serology Protocol).

We thank the reviewer for pointing out that the final cut-offs were missing in our initial submission. We added them to the description how seropositivity was determined for each infectious agent in the Supplementary Methods (section Multiplex Serology seroprevalence calculations from antigen reactivity data). These suggested ranges are also available with the online data that is publically accessible and referenced in our document

6. I think the paper can gain from the inclusion of follow up samples. However, it is not clear to me whether the found differences in seroprevalence are because of samples being close to the cutoff and are therefore ranked differently due to technical variations?

We fully agree with this reviewer (and indeed other reviewers above) that the inclusion of follow up samples is interesting. We have used these in analyses described to Reviewer #2 point 6) above and hope this reviewer also agrees with this approach. It is difficult to determine whether the observations we present are only due to fluctuation around the cutoffs as this reviewer suggests but we hope that our existing analyses noting seroconversion and seroreversion gives some insight into the minimal extent to which we would expect these effects to play a role and that our longitudinal analyses give findings that are biologically plausible.

“Identification of host-pathogen-disease relationships using a scalable Multiplex Serology platform in UK Biobank”

7. *Batch-to-batch variation (supplementary Figure 3)*

As this is supplementary, why not show it for all antigens instead of selecting only 4.

We thank the reviewer for this suggestion. We have plotted the correlations for all antigens as suggested for a comparison between two weeks as we feel that plotting all possible comparisons is excessive. We have, however, provided the raw correlation statistics for all antigens for all comparisons in Supplementary Table 4.

8. *Fig.3B slopes are missing. Choosing the same max for the x- and y-axis allows a better estimation of comparability. How comparable are the other days? When not depicting, listing of correlation coefficients might be an alternative. As the sample values look not normally distributed, a Pearson correlation might not be the best choice.*

In further addressing of this concern, we have added slopes with equations to all plots and have provided both parametric and non-parametric estimates of correlation in our assessments.

9. *Technical validation. All information on technical validation are placed in the supplementary part. For a technical validation, I would at least expect to have the intra- and inter-assay variance tested. The authors used the described screen to collect data for the inter-assay variance but do not mention intra-assay variance. How do both look like in the lower, mid and upper MFI response for the used antigens? Have they tested for the impact of freeze-thaw cycles on the antibody response? Have they tested for the impact of rheumatoid factors, HAMA, etc.?*

Intra-assay variance, i.e. intra-plate variance for Multiplex Serology has been tested and reported to be low (0.0%-3.1%, median 2.2%) before by Waterboer et al. (Clin Chem 2005).

It is known that freeze-thaw cycles may lead to degradation of proteins, such as antibodies. Thus, a differential number of freeze-thaw cycles in sera tested for one study (such as in case-control study designs) may introduce systematic bias. However, we do not expect such bias in the UKB as these ~10K sera have been treated the same before aliquoting and shipping them to DKFZ. At DKFZ, they arrived frozen and were thawed shortly before testing. We have not tested for the presence of other factors such as rheumatoid factor which, we agree, will be interesting and could be possible in the future with availability of other markers of interest through the UK Biobank study.

10. *Figure 1 is split on gender and only features on the diseases examined, yet the text appears to report the average for all samples. Please alter figure 1 or the text as such so that they are the same. Similarly, I would like to see either all diseases represented in the graph or diseases for which there are significant differences (evidence of significant between the different groups is also currently lacking, if no interactions are significant, then please state so in the figure legend) between the various cohorts (age, ethnicity and LSP). Additionally, the text for this figure only reports on higher CMV in Asian compared to White and HBV in Black compared to white individuals. Looking at the data as presented, it seems like black and Asian cohorts have higher seroprevalence than the white cohort for all diseases shown. The authors may wish to mention this in the text as it appears significant for some diseases (e.g. CMV). The authors do not also comment at any point about potential problems with the UKB. It appears to be chronically underrepresentative for all non-white ethnicities as well as for individuals with a higher degree of socio-economic deprivation, as compared to the current UK census, in addition to the stated youngest age of 40.*

“Identification of host-pathogen-disease relationships using a scalable Multiplex Serology platform in UK Biobank”

Additionally, Fig 1A and B look pretty much the same. Please check, in Fig1B should be results on males presented not on females.

We thank the reviewer for all of these comments that we have acted on. As discussed in relation to the other reviewers’ points above we have completely reworked Figure 1 in a way that we hope is more pleasant for the reader with the axis title corrected. We have also presented the relationships for all infectious agents where there is some evidence of association, where testing models allow (with sufficient numbers of individuals per groups). This was discussed in the Methods and Supplementary Methods but we have now also highlighted this in the Figure legend. Furthermore we have edited the text to separate out the descriptions by sex in line with the Figures. There is often an expectation that associations may differ by sex and since we have the power to look at these strata in this dataset we decided to stratify the presented analyses in advance of data acquisition. As per this reviewer’s request we have assessed for evidence of interaction with sex and find little evidence of this, with the exception of LSP and HSV-2 which we think is an interesting finding. These statistics are quoted in the legend as suggested by the reviewer.

Finally, the limitations of UK Biobank have been well-described and are well acknowledged by both UK Biobank lead investigators and the scientific community who utilise the cohort data (Keyes and Westreich, Lancet 393:1297) and indeed even these participation biases are shedding interesting light on the biology of human behaviour (Tyrell et al Nat Comms 12:886, and Pirastu et al Nat Genetics 53:663)). We have acknowledged this as a further limitation in our discussion (lines 413-416):

“We also acknowledge that our findings in non-White ethnicities require replication in larger studies covering a sufficient spread of different ethnicities, as the UK Biobank population is almost exclusively composed of White individuals and furthermore is not fully representative of the UK population as a whole.”

11. The number of samples listed in Supplementary Page 4 does not tally on more than one occasions. The 29 samples excluded is currently 28 based on their description (1 viscous, 8 pipetting errors, 8 incorrect dilutions and 11 insufficient bead counts). The 10110 serum samples also does not tally, as it is currently 10108 based upon their description.

We thank the reviewer for spotting this mistake and have corrected it (Supplementary Materials 5. UK Biobank Multiplex Serology quality control). There were two additional samples that had been included as paired comparator samples for blind-spiked duplicates or follow-up sampling but their paired samples were excluded as part of the described quality control and so were excluded from the final analysis.

12. Sample variance seems very high to me between days and plates. 16% plate to plate (max 23%) and 21% day to day (max 26%).

Please see our response to Reviewer 2, question 3) which provides part of the answer to this question. The control samples which we used to assess between-plate variation were selected to be collectively seropositive for as many antigens as possible, but every individual sample was seronegative for approx. 50% of all antigens. This inevitably results in high CVs for some antigens, i.e. whenever the control serum is seronegative. The reported medians largely reflect an average of assay performance where it matters (i.e., among seropositives) and where it is less relevant (i.e., among seronegatives). However, we have reported ranges of CVs across all control samples and

“Identification of host-pathogen-disease relationships using a scalable Multiplex Serology platform in UK Biobank”

antigens for transparency. Factually, and as expected, the CVs among the seropositives are much better than among the seronegatives, and below the median reported CVs.

In order to address the reviewer’s comment, we have added this as a limitation to the Discussion section (lines 406-408):

“Assay precision in this study was good when reporting combined results of seropositives (with low CVs) and seronegatives (with higher CVs); we expect this to further improve with more automated workflows.”

13. High seroprevalences in the black group are astonishing. Is this to be expected? Here it might be necessary to check the cutoffs whether they are appropriately adjusted to this ethnical group. I do understand that it will be difficult to get the respective samples, but with the view on the screening of the UKB samples clinical validation for this ethnical subgroups is recommended.

Please see our response to comment #4 regarding assay validation and cut-offs. The higher seroprevalences in the Black population are indeed an interesting finding that needs replication in suitable studies, as the UK Biobank population is almost exclusively composed of White individuals. We do not believe however that the higher seroprevalences in non-White individuals are a simple technical artefact, as the antibody responses to many of these infectious agents (e.g., HSV-2, CMV, HBV, Ct etc.) are either very low or absent in negatives, and very high in positives. There is not much of a “grey zone” for these pathogens, and even two-fold elevated backgrounds in certain ethnical groups would not generate any false-positives. To address the reviewer’s comment, we have added text about assay validation in non-White individuals, and the need for replication of our results to the Discussion section mentioned above (lines 413-416):

“We also acknowledge that our findings in non-White ethnicities require replication in larger studies covering a sufficient spread of different ethnicities, as the UK Biobank population is almost exclusively composed of White individuals.”

Minor Comments:

1. Introduction – the current introduction is light or missing on information related to either the diseases that would be examined and multiplex serology generally. I encourage the authors to add a short paragraph explaining both what diseases would be studied along with the rational behind their inclusion. I would also encourage the authors to add a short paragraph explaining why they chose to use multiplex bead assay serology and how it has been used before in similar projects.

We thank the reviewer for this comment. We have substantially expanded the Introduction in line with the comments from this and other reviewers – we hope they are in keeping with the expectations of all reviewers.

2. Supplementary Table 4 and 6 present a different number of antigens. Should be 45 antigens.

Both tables now contain 45 antigens.

3. Supplementary Fig. 4. Missing p101k

We have now included p101k in this Figure.

“Identification of host-pathogen-disease relationships using a scalable Multiplex Serology platform in UK Biobank”

4. Page 3 – *national-scale instead of prospective. It is already in progress so it cannot really be prospective.*

UK Biobank was designed in the format of a prospective cohort study to enable the sampling at baseline and linkage to assess formal association between risk factors and disease and the samples were collected to conform to this study design. We have inserted reference to this in the relevant text in the Introduction.

5. Page 3 – *reference missing for “Moreover, since many infectious agents such as Epstein-Barr virus (EBV), cytomegalovirus (CMV) and human immunodeficiency virus (HIV-1) are known to have modulatory effects on the immune system”*

We have added references to support this statement in the Introduction.

6. Page 14 – *first sentence of discussion needs a reference as it is presented as a statement of fact*

We have added references for this statement here.

7. Page 11 Supp – *figure legend need to read B, D, F and H.*

We have radically altered this Figure in light of this reviewer’s earlier comment and therefore this is no longer relevant.

Reviewers' Comments:

Reviewer #1:

Remarks to the Author:

I find the manuscript much clearer and easier to follow and to highlight the importance of this work much better.

Reviewer #2:

Remarks to the Author:

I am satisfied with the responses from the authors and am supportive of publication of the revised manuscript.

Reviewer #3:

Remarks to the Author:

I commend the authors for the substantial improvements they have made to their manuscript with this revision. It now not only reads substantially more clearly but it also provides sufficient assay and methodological details to allow investigation and replication of their work.

I have the following small comments on the revised manuscript:

Figures – It would be nice if the figures, particularly those in the main manuscript file were a little more standardised. They appear quite messy in comparison to the rest of manuscript which is very polished. Even small adjustments made in Illustrator or Inkscape to standardise the text within them would make a big improvement.

Detection antibody used – I am a little unclear of which antibody class was measured. The text says IgG and the methods suggests an IgG/IgA/IgM combination was used.

SFigure 3 – please align and scale the panels so that direct comparisons between them can be made. Currently even those using a 75 to 100 scale are not of equal size.

HHV/HCV – there are two diseases which appear to have CVs for all antigens greater than the threshold for acceptability (20%). While the results seem fine for these diseases it should be mentioned in the discussion that there were issues with their measurements, as it currently does for TG.

REVIEWERS' COMMENTS

Reviewer #1 (Remarks to the Author):

I find the manuscript much clearer and easier to follow and to highlight the importance of this work much better.

We are grateful for the positive remarks of this reviewer.

Reviewer #2 (Remarks to the Author):

I am satisfied with the responses from the authors and am supportive of publication of the revised manuscript.

We thank this reviewer for their comments.

Reviewer #3 (Remarks to the Author):

I commend the authors for the substantial improvements they have made to their manuscript with this revision. It now not only reads substantially more clearly but it also provides sufficient assay and methodological details to allow investigation and replication of their work.

We thank the reviewer for their helpful comments.

I have the following small comments on the revised manuscript:

Figures – It would be nice if the figures, particularly those in the main manuscript file were a little more standardised. They appear quite messy in comparison to the rest of manuscript which is very polished. Even small adjustments made in Illustrator or Inkscape to standardise the text within them would make a big improvement.

We have made attempts to further standardise the main Figures.

Detection antibody used – I am a little unclear of which antibody class was measured. The text says IgG and the methods suggests an IgG/IgA/IgM combination was used.

We apologise for any confusion caused here. In the analyses presented here, we have used a secondary antibody with specificity for IgG, IgM, and IgA, i.e. a mix of different antibodies targeting all 3 isotypes at once. However, the serum pre-incubation conditions we chose for the analysis contained a proprietary substance (CBS-K from Chemicon) that suppresses any IgM measurement and strongly reduces IgA antibody signals. It is well recognised that there is very little IgA present in serum and so our assay measures almost exclusively IgG. We have included two more references and have summarised this further in the Supplementary Materials text.

SFigure 3 – please align and scale the panels so that direct comparisons between them can be made. Currently even those using a 75 to 100 scale are not of equal size.

We thank the reviewer for spotting the varying x-axis scales in SFigure 3. We revised the figure by scaling x-axis limits from 70-100% for all pathogens except for T. gondii (x-axis: 40-100%) to

“Identification of host-pathogen-disease relationships using a scalable Multiplex Serology platform in UK Biobank”

improve visualization and comparability. We consider T. gondii an outlier in this figure panel, and thus did not adjust all other sub-figures to align with the T. gondii x-axis scale.

HHV/HCV – there are two diseases which appear to have CVs for all antigens greater than the threshold for acceptability (20%). While the results seem fine for these diseases it should be mentioned in the discussion that there were issues with their measurements, as it currently does for TG.

We agree with this reviewer that the statistics for HHV-6 should also be acknowledged as a limitation. This has been merged into the discussion point alongside Tg as suggested. There were so few cases of HCV that we feel any robust interpretation is difficult and therefore have not discussed this agent further in the text.